# Redundancy Undermines the Trustworthiness of Self-Interpretable GNNs

Wenxin Tai [1]   Ting Zhong [1]   Goce Trajcevski [2]   Fan Zhou [1]

## Abstract

This work presents a systematic investigation into the trustworthiness of explanations generated by self-interpretable graph neural networks (GNNs), revealing why models trained with different random seeds yield inconsistent explanations. We identify redundancy—resulting from weak conciseness constraints—as the root cause of both explanation inconsistency and its associated inaccuracy, ultimately hindering user trust and limiting GNN deployment in high-stakes applications. Our analysis demonstrates that redundancy is difficult to eliminate; however, a simple ensemble strategy can mitigate its detrimental effects. We validate our findings through extensive experiments across diverse datasets, model architectures, and self-interpretable GNN frameworks, providing a benchmark to guide future research on addressing redundancy and advancing GNN deployment in critical domains. Our code is available at https://github.com/ICDM-UESTC/TrustworthyExplanation.

## 1. Introduction

Graph neural networks (GNNs) have emerged as a powerful paradigm for learning representations on graph-structured data, serving a plethora of applications from social network analysis (Wu et al., 2022a) to molecular structure identification (Wang et al., 2022). However, notwithstanding their remarkable success, the inner workings of GNNs remain largely inscrutable. This, in turn, presents a significant barrier to their widespread adoption, especially in high-stakes domains (e.g., healthcare (Pfeifer et al., 2022), molecular dynamics (Quesado et al., 2024), finance (Rajput & Singh, 2022) and cybersecurity (Warmsley et al., 2022)) where explainability is not an option but a stringent requirement.

[1]Department of Software Engineering, University of Electronic Science and Technology of China, China [2]Department of Electrical and Computer Engineering, Iowa State University, United States. Correspondence to: Fan Zhou <fan.zhou@uestc.edu.cn>.

*Proceedings of the 42nd International Conference on Machine Learning*, Vancouver, Canada. PMLR 267, 2025. Copyright 2025 by the author(s).

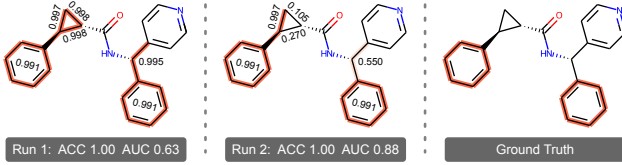

Figure 1: Illustration of explanation inconsistency of models for molecular structure detection and impactful components. Darker colors indicate higher weights.

As a result, a variety of GNN explanation methods have been developed in recent years. Some researchers focus on post-hoc methods, which interpret pre-trained GNNs using auxiliary models (Ying et al., 2019; Luo et al., 2020) or tailored strategies (Yuan et al., 2021; Chen et al., 2024b). While flexible and broadly applicable, these methods are sensitive to model initialization (Miao et al., 2022) and are affected by distribution shift (Hooker et al., 2019; Zhang et al., 2023) and, some researchers argue, may not accurately represent the true decision-making logic of the target GNN (Rudin, 2019; Ragodos et al., 2024). Such challenges have spurred a growing interest in self-interpretable GNNs (Lin et al., 2020; Dai & Wang, 2021; Miao et al., 2022; Sui et al., 2022; Deng & Shen, 2024) which, by design, jointly learn explanations and predictions.

Given the critical need for explainability in high-stakes applications, self-interpretable GNNs appear to offer a compelling solution. But do they truly live up to expectations? Motivated by experiments from (Ying et al., 2019; Zhao et al., 2023b), Figure 1 shows an example from our experiments from the domain of molecular chemistry. The left and middle portions illustrate the outcome of two runs of models trained in different random seeds aiming to detect benzene rings. Although both have same predictive accuracy (100% accuracy in detecting benzene rings), they show inconsistent explanations. Worse still, "Run 1" assigns higher weights (0.997–0.998) to an irrelevant three-membered ring as opposed to the benzene ring (0.991). Such inconsistencies and explanation-inaccuracies can yield misleading interpretations that can have dire ramifications on decision-making and affect users' trust in the practical applicability of self-interpretable GNNs.

Through systematic investigation, we identify *Redundancy* as the underlying cause: self-interpretable GNNs success-

fully identify those key features but also overemphasize some irrelevant ones. We argue that hyperparameters used to enforce explanation conciseness are set overly relaxed, granting the model excessive flexibility to retain irrelevant features, thus weakening the trustworthiness of GNN explanations. Through theoretical analysis and experiments, we show that redundancy is inherently difficult to eliminate. Existing techniques (Madhyastha & Jain, 2019; Zhao et al., 2023a; Deng & Shen, 2024), despite promising results in other related fields, fail to address this challenge.

While a complete solution to eliminating redundancy remains an open challenge, we find that a simple, tuning-free strategy—Explanation Ensemble (EE)—can significantly mitigate its adverse effects. By aggregating explanations from multiple independently trained models, EE retains the signal of truly relevant features while suppressing the noise from irrelevant ones, leading to more consistent and accurate explanations. Extensive experiments across diverse datasets, model architectures, and self-interpretable GNN frameworks validate the effectiveness of EE.

In sum, we present a systematic investigation into the trustworthiness of explanations generated by self-interpretable GNNs. Our central contribution is identifying redundancy as a fundamental cause of explanation inconsistency and inaccuracy. To address this, we introduce EE as both a practical mitigation strategy and a benchmark for future research. We hope our findings will encourage the development of more principled, theoretically grounded approaches, paving the way for the safe and effective deployment of GNNs in high-stakes applications.

## 2. Preliminaries

We begin by formally defining graphs and GNNs. Following this, we define self-interpretable GNNs and classify existing methods into four types based on their design principles.

### 2.1. Basic Definitions

**Graph.** A graph $G$ is a quadruplet $(\mathcal{V}, \mathcal{E}, \mathbf{X}, \mathbf{A})$, where $\mathcal{V} = \{v_1, v_2, \ldots, v_N\}$ is the set of nodes, $\mathcal{E} \subseteq \mathcal{V} \times \mathcal{V}$ is the set of edges, $\mathbf{X} \in \mathbb{R}^{N \times d}$ is the feature matrix where each row represents the feature vector of a node, and $\mathbf{A} \in \{0, 1\}^{N \times N}$ is the adjacency matrix, where $\mathbf{A}_{ij} = 1$ if there is an edge between nodes $v_i$ and $v_j$, and 0 otherwise.

**Graph Neural Network (GNN).** GNN (Scarselli et al., 2008) is a type of a neural network designed to operate on graph-structured data, making it particularly well-suited for tasks where understanding relationships between entities is crucial. Taking the graph classification task as an example, we structure the GNN as a combination of two modules: $f = h_{\hat{Y}} \circ h_Z$. In this context, the module $h_Z$ ($G \mapsto \mathbb{R}^d$) learns the graph representation, while $h_{\hat{Y}}$ ($\mathbb{R}^d \mapsto \mathbb{R}$) gener-

ates the prediction $\hat{Y}$ to approximate $Y$.

**Self-Interpretable GNNs.** To enhance the interpretability of GNNs, current explanation methods aim to illuminate the model's decision-making process by identifying a concise yet crucial set of features within the graph. Therefore, a self-interpretable GNN can be represented as $f = h_{\hat{Y}} \circ h_Z \circ h_{G_s}$, and the optimization objective is formalized as:

$$\max_{G_s \subseteq G, |G_s| \leq K} I(G_s; Y) \tag{1}$$

where $G$, $G_s$, and $Y$ represent the graph, its subset, and the graph label, respectively. The goal is to maximize the mutual information (MI) between the graph subset and the label while controlling the size ($K$) of the subset. Along the lines of prevalent graph explanation efforts (Amara et al., 2022; Chen et al., 2023), we focus on the contribution of the structural features, namely the edges.

### 2.2. Taxonomy of Self-Interpretable GNNs

We classify existing self-interpretable GNNs into four types based on their design principles.

**Type I (Attention).** The attention mechanism, introduced in Vaswani et al. (2017), is an early method for understanding how NNs make decisions. It works by assigning different levels of importance to different features. Attention-based self-interpretable GNNs take advantage of this built-in ability to learn which edges are most important during training. This happens without the need for any extra rules or penalties (loss constraints) to force the model to explain itself (Velickovic et al., 2018):

$$\mathcal{L}_{\text{GE}} = \mathcal{L}_{\text{CE}}(Y, \hat{Y}|G_s) \tag{2}$$

where $\mathcal{L}_{\text{CE}}$ is classification loss (e.g., cross-entropy loss).

**Type II (Causal Learning).** Some works approach GNN interpretability from a causal perspective, aiming at learning truly causal patterns that go beyond pure spuriosity-prone statistical correlation (Wu et al., 2022b). Techniques such as disentanglement and causal intervention are often used to optimize the learning process (Sui et al., 2022):

$$\mathcal{L}_{\text{GE}} = \mathcal{L}_{\text{CE}}(Y, \hat{Y}|G_s) + \beta \cdot \mathbb{D}_{\text{KL}}(\mathbb{P}_{\boldsymbol{\theta}}(\bar{Y}|\bar{G}_s)\|\mathbb{Q}(\bar{Y}))$$
$$+ \gamma \cdot \mathcal{L}_{\text{CE}}(Y, \hat{Y}'|G_s \cup \bar{G}_s') \tag{3}$$

where $\beta$ and $\gamma$ are pre-defined hyperparameters, $\bar{G}_s = G \setminus G_s$ is the complement of $G_s$, and $\bar{G}_s'$ represents the result of intervening on $\bar{G}_s$ (e.g., shuffling the edge weights of $\bar{G}_s$). $\mathbb{Q}(\bar{Y})$ is often set to a uniform distribution.

**Type III (Size Constraint).** To encourage more concise and human-understandable explanations, some works extend attention-based methods by adding regularization terms that constrain the size of $G_s$. A representative example is

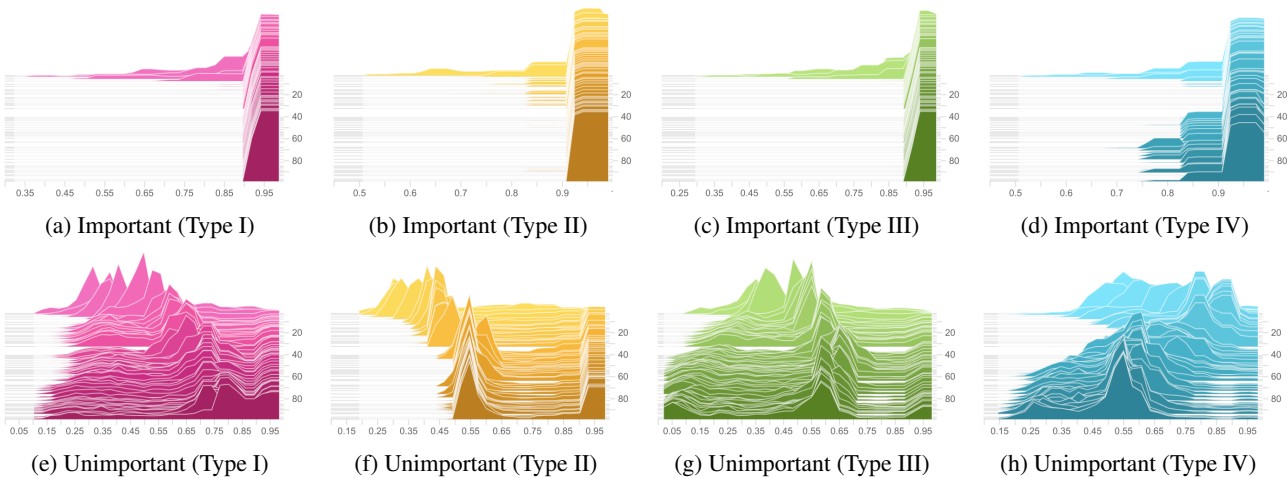

Figure 2: Histograms of edge weights trained on the BA-2MOTIFS dataset. The first row shows the distribution of edge weights assigned to truly important edges, while the second row shows those assigned to unimportant edges. The vertical axis represents the epoch numbers. Histograms for other datasets are provided in Appendix B.

Lin et al. (2020), which introduces a sparsity loss that normalizes the constraint by dividing it by the graph size:

$$\mathcal{L}_{\text{GE}} = \mathcal{L}_{\text{CE}}(Y, \hat{Y}|G_s) + \beta \cdot \frac{|G_s|}{|G|} \qquad (4)$$

**Type IV (MI Constraint).** Some works constrain the size of $G_s$ from an MI perspective, encouraging concise explanations by minimizing MI between $G_s$ and $G$. One work (Yu et al., 2021) uses a computationally expensive technique (Belghazi et al., 2018) to estimate MI, while the other (Miao et al., 2022) adopts a tractable variational upper bound:

$$\mathcal{L}_{\text{GE}} = \mathcal{L}_{\text{CE}}(Y, \hat{Y}|G_s) + \beta \cdot \mathbb{D}_{\text{KL}}(\mathbb{P}_{\boldsymbol{\theta}}(G_s|G)||\mathbb{Q}(G_s)) \quad (5)$$

where $\mathbb{Q}(G_s)$ is a Bernoulli distribution. Since the sampling process is non-differentiable, Gumbel-Softmax trick (Jang et al., 2017) is employed for continuous relaxation.

## 3. Inconsistency Investigation

We begin by examining two commonly presumed sources of explanations inconsistency—(1) training instability (Madhyastha & Jain, 2019), and (2) spurious correlations (Zhao et al., 2023a;b)—both of which have been studied as contributing factors in previous literature.

### 3.1. Training Instability

Training instability has been identified as a key contributor to explanation inconsistency in attention-based NNs for NLP tasks (Madhyastha & Jain, 2019). Their work suggests that the inherent complexity of non-convex loss surfaces—characterized by numerous saddle points and local minima—hampers stable training, resulting in inconsistent

model performance, which in turn can manifest as variation in the generated explanations. To mitigate this issue, Madhyastha & Jain (2019) adopted Stochastic Weight Averaging (SWA) (Izmailov et al., 2018) and its variants, which stabilize training by averaging weights collected from different points along the optimization trajectory. This approach encourages convergence to flatter optima, thereby improving explanation consistency.

To investigate whether training instability leads to explanation inconsistency in self-interpretable GNNs, we apply SWA to four self-interpretable GNN frameworks across four datasets. Our experimental results (cf. Table 1) show that SWA improves explanation consistency in only 7 out of 16 tests. This suggests that while training instability may contribute to explanation inconsistency, it does not fully explain the phenomenon in self-interpretable GNNs.

### 3.2. Spurious Correlations

Spurious correlations have been identified as a key contributor to explanation inconsistency in post-hoc GNN methods (Zhao et al., 2023a;b). Their works argue that when the explainer is optimized solely based on predicted outputs—e.g., by training it to generate subgraphs whose predictions align with those of raw graphs—it becomes prone to overfitting spurious patterns: features correlated with the output but not causally informative. This vulnerability stems from the two-stage pipeline, where the explainer is trained separately to mimic the behavior of a pre-trained GNN and lacks access to its training dynamics and inductive biases. Depending on initialization and optimization dynamics, the explainer may attend to different features across runs, sometimes capturing truly relevant ones and other times latching onto spurious cues. Such instability results in explanation

inconsistency across random seeds.

To examine whether spurious correlations underlie explanation inconsistency in self-interpretable GNNs, we conduct experiments on several benchmark datasets. As shown in Figure 2 (a)–(d), when evaluated on synthetic datasets without deliberately introduced spurious patterns, self-interpretable GNNs consistently assign high weights to truly important edges, indicating strong robustness to spurious correlations. This likely stems from the nature of explanation generation: unlike post-hoc methods, which mimic the behavior of pre-trained GNNs and are thus susceptible to optimization bias (Miao et al., 2022) and confounding effects (Wu et al., 2022b), self-interpretable GNNs learn explanations jointly with predictions, embedding them directly into the model's decision-making process.

## 4. Redundancy in Explanation

Figure 2 (e)–(h) suggests another source of explanation inconsistency in self-interpretable GNNs: redundancy. When trained with different random seeds, models consistently identify key features but vary in their inclusion of irrelevant ones, leading to inconsistent explanations. We attribute this redundancy to the limitations of global hyperparameters in enforcing conciseness. Given the varying graph sizes and explanation sizes across instances, fixed hyperparameter values are rarely optimal, allowing the model excessive flexibility to retain redundant edges.

The suboptimality of global hyperparameters is further illustrated in Figure 3, where histograms of $|G_s|/|G|$ show that the generated explanations consistently include more edges than the ground-truth ones. This behavior is expected: in the absence of ground-truth explanations in real-world scenarios, hyperparameters are empirically tuned to balance classification accuracy and explanation conciseness. In our experiments, retaining more edges stabilizes training and improves classification performance. This also aligns with recent findings (Wu et al., 2022b; Miao et al., 2022), which recommend retaining 50%–80% of edges to achieve a favorable trade-off. To formalize the notion of redundancy, we establish a connection to explanation budgets as follows:

**Proposition 4.1.** *Define the crucial graph subset as the optimal GNN explanation $G_s^*$. When $K \geq |G_s^*|$, there exist $\geq 1$ valid GNN explanations that can satisfy Equation* (1).

The proof is provided in Appendix A. As discussed above, because the true size of graph explanations is unknown in practice, designing size-specific loss objectives to directly address redundancy is challenging.

Instead of explicitly targeting explanation size, recent work (Deng & Shen, 2024) adopted contrastive learning (Oord et al., 2018) as an indirect strategy to eliminate redundancy. Their method is grounded in two key properties that an ideal

GNN explanation $G_s^*$ should satisfy: (1) *Sufficiency*, meaning the explanation must preserve all critical information required for accurate prediction; and (2) *Necessity*, meaning the removal of any edge in $G_s^*$ would impair the model's prediction due to loss of crucial information.

To enforce these properties, the authors introduced an auxiliary contrastive loss. Given a generated explanation $G_s$ (serving as the anchor), they constructed two variants: (1) a positive sample $G_s^+$, obtained by adding edges to $G_s$ such that it still contains sufficient information for accurate prediction; and (2) a negative sample $G_s^-$, obtained by removing essential edges, thereby disrupting critical substructures and causing the GNN to fail in making correct predictions. The contrastive loss is formulated as[1]:

$$\mathcal{L}_{\text{CL}} = -\log \frac{\exp\left(\text{sim}(\hat{\mathbf{z}}_s, \hat{\mathbf{z}}_s^+)/\tau\right)}{\exp\left(\text{sim}(\hat{\mathbf{z}}_s, \hat{\mathbf{z}}_s^+)/\tau\right) + \exp\left(\text{sim}(\hat{\mathbf{z}}_s, \hat{\mathbf{z}}_s^-)/\tau\right)}$$

where $\hat{\mathbf{z}}_s$, $\hat{\mathbf{z}}_s^+$, and $\hat{\mathbf{z}}_s^-$ denote the projected representations of $G_s$, $G_s^+$, and $G_s^-$, respectively; for example, $\hat{\mathbf{z}}_s$ is obtained by applying an MLP to $\mathbf{z}_s$, i.e., $\hat{\mathbf{z}}_s = \text{MLP}(\mathbf{z}_s)$. The function $\text{sim}(\cdot, \cdot)$ denotes cosine similarity, and $\tau$ is a temperature hyperparameter. This loss encourages maximizing the similarity between the representations of $G_s$ and $G_s^+$ while minimizing that between $G_s$ and $G_s^-$. As a result, the model learns to generate concise and meaningful explanations that satisfy both sufficiency and necessity, reducing redundancy without sacrificing predictive power.

However, it is important to note that the efficacy of the above strategy relies on a restrictive assumption: for each graph $G$, the optimal explanation $G_s^*$ should simultaneously satisfy sufficiency and necessity. In practice, this assumption is often unattainable. For instance, in datasets like BENZENE (Morris et al., 2020), where graphs are labeled to indicate the presence of one or more benzene rings, each independent explanation (i.e., each benzene ring) constitutes a valid explanation. In such cases, sufficiency and necessity represent inherently conflicting criteria: prioritizing sufficiency leads to retaining all benzene rings, whereas emphasizing necessity discards alternative valid substructures, as a single benzene ring suffices to explain the graph. This conflict can be formalized as follows:

**Proposition 4.2.** *Assume that $G$ has multiple independent explanations $G_s^* = \{G_s^{1,*}, G_s^{2,*}, \ldots, G_s^{m,*}\}$ with pairwise disjoint edge sets. Then, no single edge is strictly necessary.*

The proof is provided in Appendix A. To investigate the practical implications, we conduct experiments on two datasets: BA-2MOTIFS (Luo et al., 2020) and BENZENE (Sanchez-Lengeling et al., 2020), which exemplify single-explanation and multi-explanation scenarios, respectively.

Our quantitative results (see Table 2 in Appendix D) confirm

---

[1]We use one positive and one negative sample for simplicity.

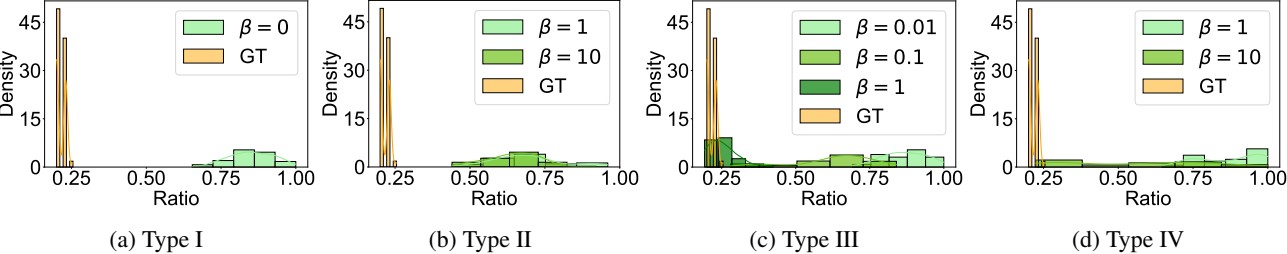

Figure 3: Histograms of $|G_s|/|G|$ on the BA-2MOTIFS dataset. GT refers to Ground Truth. Only results from settings that successfully converge are shown. Histograms for other datasets are provided in Appendix C.

the theoretical insights presented above. Specifically, on the BA-2MOTIFS dataset, incorporating contrastive loss reduces explanation inconsistency and improves accuracy, as the removal of necessary edges directly disrupts the unique explanation. In contrast, on the BENZENE dataset, removal of certain edges does not invalidate the remaining explanations. Consequently, both positive and negative samples retain sufficient information for accurate predictions, resulting in ambiguity that leads to degraded performance across all metrics due to convergence difficulties. These findings highlight the need for broadly applicable methods that do not rely on restrictive dataset-specific assumptions.

## 5. Explanation Ensemble

In this section, we demonstrate that simple Explanation Ensemble (EE), which aggregates explanations from multiple independently trained models, can effectively alleviate the detrimental effects of explanation redundancy. To formally analyze the effectiveness of EE in reducing explanation inconsistency, we present the following proposition:

**Proposition 5.1.** *Let $X$ be a random variable with $\mathbb{E}[X] = a$ and $\mathrm{Var}(X) = b$, where $X \in [0, 1]$. Let $x_1, x_2$ be independent draws from $X$, and let $\bar{X}_1, \bar{X}_2$ be the sample means of two independent $n$-samples from $X$. For a positive hyperparameter $k \in (0, 1]$, we have:*

$$\mathbb{P}(|\bar{X}_1 - \bar{X}_2| < k|x_1 - x_2|) \geq 1 - \frac{2b}{nk^2|x_1 - x_2|^2} \quad (6)$$

The proof is provided in Appendix A. In self-interpretable GNNs, $X$ represents the importance score (ranging from 0 to 1) assigned to an edge, reflecting its contribution to the model's prediction. The inconsistency in edge importance between two models is quantified by $|x_1 - x_2|$, where $x_1$ and $x_2$ are the importance scores assigned to the same edge. Proposition 5.1 shows that by aggregating explanations from $n$ independently trained models (forming two ensembles with average importance scores $\bar{X}_1$ and $\bar{X}_2$), the probability that the ensemble inconsistency $|\bar{X}_1 - \bar{X}_2|$ is less than $k$ times the individual model inconsistency $|x_1 - x_2|$ has a lower bound that increases with $n$. Note that the above edge-level analysis readily extends to subgraphs, as graph-level

inconsistency can be viewed as the average inconsistency of its constituent edges.

To formally analyze the effectiveness of EE in improving explanation accuracy, we present the following proposition:

**Proposition 5.2.** *Let $X$ and $W$ be independent random variables with $X \in [0, 1]$ and $W \in [\delta, 1]$, where $0 < \delta \leq 1$. Let $\bar{X}$ be the sample mean of $n$ independent observations of $X$. If $\mathbb{E}[X] < \delta$, then we have:*

$$\mathbb{P}(\bar{X} < W) \geq 1 - \exp(-2n(\delta - \mathbb{E}[X])^2) \quad (7)$$

The proof is provided in Appendix A. Let $X$ and $W$ denote the importance scores of an irrelevant and a relevant edge, respectively, where $X \in [0, 1]$ and $W \in [\delta, 1]$ with $0 < \delta \leq 1$[2]. Proposition 5.2 demonstrates that aggregating explanations from $n$ independently trained models, yielding the average importance score $\bar{X}$ for the irrelevant edge, increases a lower bound on the probability of correctly distinguishing irrelevant edges from relevant ones (i.e., $\mathbb{P}(\bar{X} < W)$). This increased probability directly contributes to a higher ROC-AUC score, a key metric for evaluating explanation accuracy (Ying et al., 2019).

From a more intuitive perspective, the explanation generated by a self-interpretable GNN can be viewed as consisting of two components: (1) edges that the model genuinely deems important, and (2) edges that are assigned high importance just because sufficient budget allocation (redundancy). EE can improve explanation consistency and accuracy, as these redundant edges typically exhibit high variance and tend to receive lower average importance after aggregation.

## 6. Experiments

**Metrics.** To evaluate explanation consistency, we follow Zhao et al. (2023a;b), run each method multiple times with different random seeds and report the average Structural Hamming Distance (SHD) (Tsamardinos et al., 2006) across the generated explanations. SHD quantifies the structural differences between two graphs by counting mismatched

---

[2]We assume that truly important edges are assigned high importance scores, an assumption supported by our earlier investigations.

Table 1: We run each method 10 times, reporting the mean and standard deviation. Results significantly outperforming the baseline ✗ (GIN, paired t-test, $p < 0.05$) are underlined. Values in brackets show performance changes after Explanation Ensemble. Dark green indicates performance improvement, while dark red indicates performance degeneration.

| METHOD | METRIC | STRATEGY | BA-2MOTIFS | 3MR | BENZENE | MUTAGENICITY |
|---|---|---|---|---|---|---|
| **Type I** (Attention) | SHD | ✗ | 4.75±4.68 (↓ **3.53**) | 5.59±3.19 (↓ **2.68**) | 6.11±4.34 (↓ **3.21**) | 5.02±6.60 (↓ **4.99**) |
| | | +SWA | 12.07±8.21 (↓ **1.81**) | 4.27±2.56 (↓ **2.34**) | 7.42±4.61 (↓ **2.55**) | 11.24±6.18 (↓ **5.64**) |
| | | +EA | 4.04±3.80 (↓ **2.49**) | 9.79±6.56 (↓ **5.04**) | 2.01±2.63 (↓ **1.96**) | 9.63±14.37 (↓ **9.10**) |
| | AUC (%) | ✗ | 99.30±0.34 (↑ **0.31**) | 97.25±0.66 (↑ **1.27**) | 83.50±2.23 (↑ **7.40**) | 91.71±6.00 (↑ **6.25**) |
| | | +SWA | 99.03±0.36 (↑ **0.11**) | 97.81±0.13 (↑ **0.73**) | 85.78±1.48 (↑ **4.26**) | 98.57±0.34 (↑ **0.66**) |
| | | +EA | 97.73±1.41 (↑ **1.31**) | 97.86±0.75 (↑ **1.30**) | 88.61±3.76 (↑ **5.11**) | 97.19±3.09 (↑ **2.23**) |
| | ACC (%) | ✗ | 97.10±8.70 (↑ **2.90**) | 96.53±1.45 (↑ **0.70**) | 91.60±0.66 (↑ **1.15**) | 92.97±0.78 (↑ **0.94**) |
| | | +SWA | 98.20±1.53 (↑ **0.80**) | 95.32±0.99 (↑ **0.87**) | 89.63±2.02 (↑ **1.20**) | 89.59±1.53 (↑ **0.61**) |
| | | +EA | 100.00±0.00 (**0.00**) | 97.99±0.70 (↑ **1.31**) | 90.71±0.66 (↑ **0.70**) | 90.33±1.13 (↑ **1.89**) |
| **Type II** (Causal Learning) | SHD | ✗ | 7.75±5.11 (↓ **3.51**) | 5.92±3.79 (↓ **2.98**) | 9.17±5.79 (↓ **4.07**) | 19.57±14.49 (↓ **10.44**) |
| | | +SWA | 11.29±7.58 (↓ **6.59**) | 4.01±2.60 (↓ **1.97**) | 9.02±5.72 (↓ **4.71**) | 12.60±6.21 (↓ **4.88**) |
| | | +EA | 7.59±9.63 (↓ **7.57**) | 9.24±4.83 (↓ **3.85**) | 4.14±4.51 (↓ **2.49**) | 19.11±16.31 (↓ **10.45**) |
| | AUC (%) | ✗ | 98.63±1.33 (↑ **0.99**) | 96.24±1.58 (↑ **2.08**) | 77.64±3.05 (↑ **8.20**) | 96.39±1.38 (↑ **2.83**) |
| | | +SWA | 95.93±1.99 (↑ **2.72**) | 96.56±0.31 (↑ **1.48**) | 78.80±2.99 (↑ **5.66**) | 98.32±0.40 (↑ **0.98**) |
| | | +EA | 96.56±1.95 (↑ **2.68**) | 96.46±1.55 (↑ **2.08**) | 83.69±4.02 (↑ **8.13**) | 95.56±3.22 (↑ **3.35**) |
| | ACC (%) | ✗ | 91.89±11.68 (↑ **6.11**) | 94.22±2.10 (↑ **1.97**) | 84.30±5.65 (↑ **3.28**) | 91.21±1.36 (↑ **1.01**) |
| | | +SWA | 81.30±12.10 (↑ **9.70**) | 93.39±1.48 (↑ **1.07**) | 84.33±4.84 (↑ **2.50**) | 89.05±0.89 (↓ **0.54**) |
| | | +EA | 99.90±0.30 (↑ **0.10**) | 97.30±1.35 (↑ **0.94**) | 87.83±1.18 (↑ **1.33**) | 89.93±0.92 (↑ **0.27**) |
| **Type III** (Size Constraint) | SHD | ✗ | 11.41±9.37 (↓ **6.93**) | 3.50±2.44 (↓ **1.82**) | 6.91±4.48 (↓ **2.47**) | 5.97±4.22 (↓ **2.40**) |
| | | +SWA | 11.84±8.57 (↓ **2.85**) | 3.85±2.43 (↓ **1.90**) | 7.58±4.63 (↓ **4.59**) | 7.23±4.13 (↓ **3.78**) |
| | | +EA | 9.64±8.66 (↓ **7.47**) | 7.17±4.09 (↓ **3.17**) | 16.34±9.80 (↓ **2.15**) | 15.17±9.86 (↓ **4.22**) |
| | AUC (%) | ✗ | 99.32±0.36 (↑ **0.35**) | 96.99±0.76 (↑ **1.43**) | 84.38±2.71 (↑ **8.09**) | 98.11±0.38 (↑ **1.01**) |
| | | +SWA | 99.19±0.33 (↑ **0.09**) | 97.60±0.22 (↑ **0.81**) | 85.67±2.02 (↑ **5.16**) | 98.16±0.43 (↑ **1.12**) |
| | | +EA | 96.54±4.43 (↑ **1.93**) | 97.04±1.37 (↑ **2.13**) | 85.72±5.35 (↑ **7.27**) | 97.96±0.87 (↑ **1.47**) |
| | ACC (%) | ✗ | 95.50±12.51 (↑ **4.50**) | 97.30±1.31 (↑ **0.62**) | 91.11±0.62 (↑ **1.22**) | 89.52±0.99 (↑ **1.69**) |
| | | +SWA | 97.10±2.66 (↑ **0.90**) | 94.25±2.71 (↑ **1.25**) | 88.24±3.36 (↑ **1.34**) | 88.51±1.40 (↑ **0.34**) |
| | | +EA | 99.80±0.40 (↑ **0.20**) | 97.43±0.72 (↑ **1.53**) | 89.23±1.92 (↑ **1.93**) | 89.22±0.91 (↑ **1.32**) |
| **Type IV** (MI Constraint) | SHD | ✗ | 3.29±2.97 (↓ **1.74**) | 7.71±5.51 (↓ **3.69**) | 4.13±3.15 (↓ **2.50**) | 13.30±10.29 (↓ **7.17**) |
| | | +SWA | 1.63±1.47 (↓ **0.65**) | 4.97±3.65 (↓ **2.95**) | 5.81±3.98 (↓ **2.90**) | 0.14±0.24 (↓ **0.14**) |
| | | +EA | 0.54±1.08 (↓ **0.54**) | 3.49±2.50 (↓ **1.86**) | 6.73±6.29 (↓ **3.45**) | 17.02±18.09 (↓ **9.31**) |
| | AUC (%) | ✗ | 98.44±0.60 (↑ **0.32**) | 98.37±0.31 (↑ **0.80**) | 90.66±0.88 (↑ **2.00**) | 99.00±0.30 (↑ **0.39**) |
| | | +SWA | 98.53±0.25 (↑ **0.06**) | 98.97±0.10 (↑ **0.29**) | 91.10±0.39 (↑ **1.14**) | 99.07±0.31 (↑ **0.29**) |
| | | +EA | 96.89±1.67 (↑ **1.54**) | 98.74±0.22 (↑ **0.52**) | 92.40±0.86 (↑ **1.88**) | 99.23±0.23 (↑ **0.26**) |
| | ACC (%) | ✗ | 100.00±0.00 (**0.00**) | 98.54±0.80 (↑ **0.76**) | 91.48±0.87 (↑ **0.85**) | 92.43±1.00 (↑ **0.13**) |
| | | +SWA | 100.00±0.00 (**0.00**) | 98.13±0.46 (↑ **0.83**) | 91.48±0.49 (↑ **0.93**) | 91.38±0.94 (↑ **0.84**) |
| | | +EA | 99.60±0.66 (↑ **0.40**) | 97.92±0.84 (↑ **0.34**) | 88.80±0.39 (↓ **0.05**) | 89.32±1.26 (↑ **1.22**) |

edges (either missing or extra edges). Since NNs output continuous soft edge weights, these must first be discretized to compute SHD. A threshold of 0.5 is used in this study: edges with weight greater than 0.5 are considered present, while those below 0.5 are deemed absent[3]. To evaluate explanation accuracy, we choose datasets with ground-truth explanations, as they provide objective benchmarks for comparison (Chen et al., 2024a). Following established practices (Wu et al., 2022b; Miao et al., 2022), we use ROC-AUC (AUC) to evaluate explanation quality and Accuracy (ACC) to evaluate predictive performance.

**Datasets.** We select four datasets: a synthetic dataset, BA-

2MOTIFS (Luo et al., 2020), and three real-world dataset – 3MR (Rao et al., 2022), BENZENE (Sanchez-Lengeling et al., 2020), and MUTAGENICITY (Debnath et al., 1991) – all sourced from the graph learning community and have ground-truth explanation labels.A detailed discussion on the choice of metrics and datasets is provided in Appendix E.

**Baselines.** To ensure fair, reliable, and comprehensive experiments, we evaluate two GNN backbones, GIN (Xu et al., 2019) and GCN (Kipf & Welling, 2017), alongside four types of self-interpretable GNN methods: Type I to Type IV. Due to space limitations, GIN results are presented in the main paper, with GCN results in Appendix F.

**+SWA:** Madhyastha & Jain (2019) introduced SWA to address attention inconsistency. To investigate whether training instability contributes to explanation inconsistency of self-interpretable GNNs, we use SWA as a baseline.

---

[3]We chose a threshold of 0.5 because determining whether an edge is important is essentially a binary classification task, and 0.5 is a standard and prior-free choice. In contrast, other thresholds or Top-K selections all require domain knowledge.

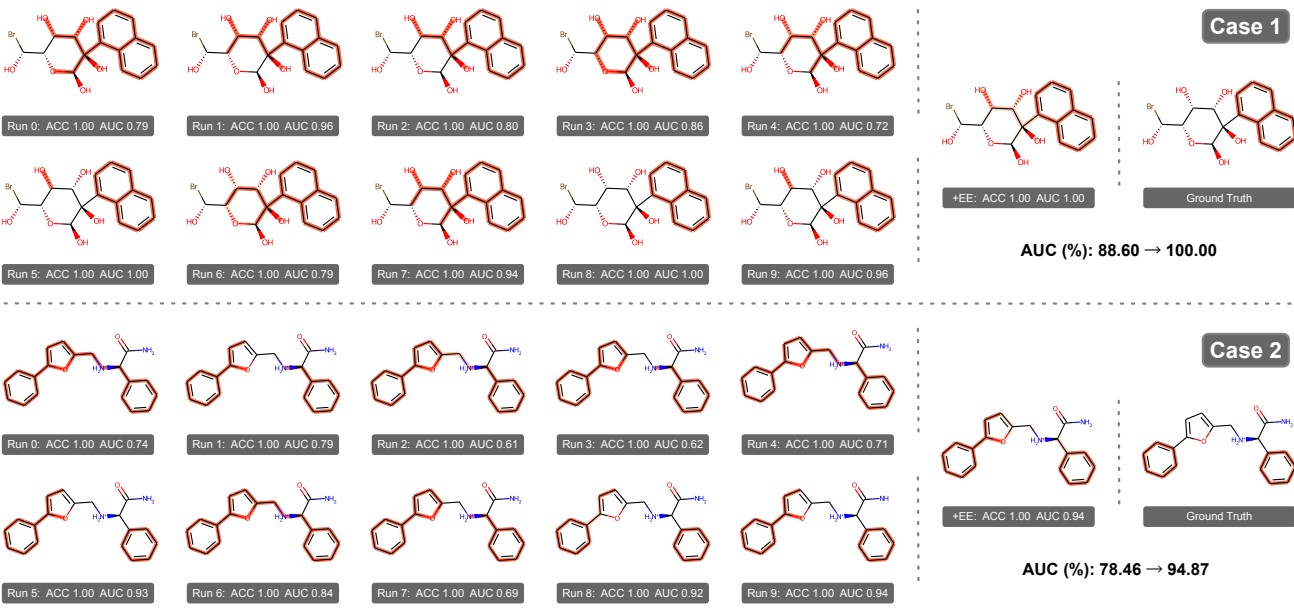

Figure 4: Illustration of Explanation Ensemble. Darker colors indicate higher weights. Only edges with weights greater than 0.5 are highlighted, as they collectively form the explanations.

**+EA:** Zhao et al. (2023a;b) argued that optimization solely based on predicted labels can easily lead to overfitting to spurious correlations. To address this, they proposed several alignment-based loss objectives to enhance semantic consistency between raw graphs and identified subgraphs in the embedding space. An example implementation employs the Euclidean absolute distance to align graph representations. Formally, let $\mathbf{z}$ and $\mathbf{z}_s$ denote the pooled embeddings of the raw graph and the identified key graph subset, respectively. The embedding alignment (EA) loss is defined as:

$$\mathcal{L}_{\text{EA}} = \|\mathbf{z} - \mathbf{z}_s\|_2. \tag{8}$$

We use EA as a baseline. Additional details about experimental setup are provided in Appendix G.

### 6.1. Overall Performance

Table 1 reports the overall performance and we have the following observations:

**(O1):** SWA does not consistently improve the quality of GNN explanations: it decreases SHD in approximately 44% of cases and increases AUC in around 63%. This result can be attributed to two factors. First, self-interpretable GNNs often train stably and converge to near-optimal solutions, leaving limited room for further gains. Second, averaging weights may pull the model away from a sharp yet accurate optimum toward a flatter but suboptimal region in the loss landscape. While flatter minima can sometimes improve generalization, they may also lead to less precise explanations, thereby reducing performance in some cases.

**(O2):** EA also fails to improve the quality of GNN explana-

tions. For example, on the BA-2MOTIFS dataset, it causes an AUC drop of up to 3% across all self-interpretable GNN frameworks. We attribute this to the difference in how the alignment loss operates in different settings. In post-hoc methods, the alignment loss is applied solely to the explainer, which learns to match the representation of a fixed, pre-trained GNN. Since the target representation remains unchanged across runs, this alignment directly promotes explanation consistency. In contrast, self-interpretable GNNs train both the explainer and the GNN jointly, and the loss influences both. As a result, the explainer is constantly aligning to a moving target—its own co-evolving GNN—which weakens the intended effect of enforcing consistency.

**(O3):** EE is the only approach that consistently improves explanation quality across all evaluated settings, reducing SHD and improving AUC in all 48 cases. Moreover, EE also improves downstream classification accuracy (ACC) in 46 out of 48 cases. The simplicity of EE and its independence from specific training procedures (unlike EA, which requires an additional loss term, and SWA, which necessitates changes to the training regime) make it highly suitable for real-world applications.

### 6.2. Case Study

While Table 1 provides a quantitative overview of EE's performance, we now present two case studies to qualitatively illustrate how EE impacts explanation accuracy (Figure 4).

In the first case, the average AUC of the initial explanations is 88.60%. After applying EE, the weights of irrelevant edges are effectively reduced, leading to a perfect AUC of

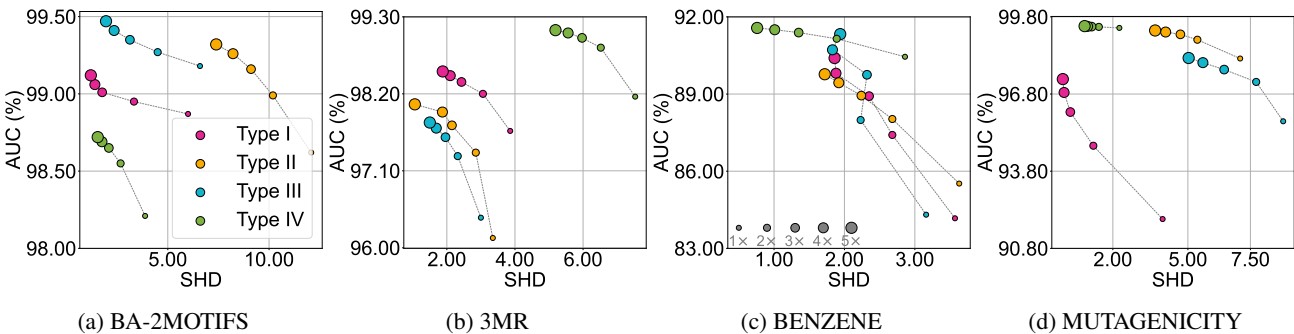

Figure 5: Trustworthiness (SHD and AUC) *vs.* Efficiency ($1\times$, $2\times$, $3\times$, $4\times$, $5\times$) on four datasets.

100%. This demonstrates EE's ability to retain the signal of truly relevant features while suppressing the noise from irrelevant ones. In the second case, the average AUC of the initial explanations is 78.46%. Applying EE substantially improves the AUC to 87%, with the weights of most irrelevant edges significantly reduced. However, a few irrelevant edges still retain relatively high weights because these edges are consistently identified as important across all 10 models. This observation also aligns with our theoretical analysis in Proposition 5.2, which suggests that EE's effectiveness is diminished when the weights of certain irrelevant edges remain consistently high.

Together, these two cases demonstrate EE's capability to improve explanation accuracy. Given its simplicity, empirical effectiveness, and theoretical foundation, we believe that EE can serve as a strong baseline for future research on trustworthy self-interpretable GNNs and may inspire the development of more principled solutions.

### 6.3. Efficiency-Trustworthiness Trade-off

Although EE consistently improves explanation quality, its ensemble nature requires training multiple models, increasing training and deployment costs linearly with $n$. To investigate the relationship between trustworthiness and efficiency, we vary $n$ and plot the resulting SHD and AUC values. As shown in Figure 5, we have the following two key observations: (1) Increasing $n$ generally leads to improved performance across all metrics and datasets. (2) Most significant performance gain is observed when $n$ is increased from 1 to 2. These findings suggest $n = 2$ as a practical choice in balancing trustworthiness and efficiency. Another limitation of EE—its incompatibility with faithfulness metrics—is discussed in Appendix H.

## 7. Related Work

A variety of methods have been proposed to explain GNN predictions, with most efforts focusing on improving explanation accuracy (Ying et al., 2019; Yuan et al., 2021; Wang et al., 2021; Dai & Wang, 2021) or computational efficiency

(Luo et al., 2020; Lu et al., 2024; Luo et al., 2024). However, as GNNs are increasingly deployed in high-stakes domains, other critical properties – such as robustness (Bajaj et al., 2021; Li et al., 2024; Fang et al., 2024), generalizability (Wu et al., 2022b; Miao et al., 2022; Azzolin et al., 2025), and fairness (Medda et al., 2024; Dong et al., 2022) – have gained growing attention (Dai et al., 2024).

Zhao et al. (2023a;b) took a pivotal step in investigating the consistency of GNN explanations. They concluded that spurious correlations are the core reason behind inconsistency in post-hoc GNN explanation methods (Ying et al., 2019; Luo et al., 2020). Inspired by their work, we conduct an extended investigation on self-interpretable GNNs (Velickovic et al., 2018; Sui et al., 2022; Miao et al., 2022; Deng & Shen, 2024), and find that explanation inconsistency persists even in the absence of spurious correlations. This observation motivates us to look beyond existing hypotheses, ultimately leading to the identification of redundancy—stemming from the weak conciseness constraints—as a core and previously underexplored reason for inconsistency.

## 8. Conclusion & Future Work

In this work, we identify redundancy as a root cause of explanation inconsistency and its associated inaccuracy in self-interpretable GNNs. While existing methods struggle to eliminate this redundancy, we demonstrate that a simple, tuning-free ensemble strategy EE can substantially mitigate its negative effects, leading to more consistent and accurate explanations. Extensive experiments across diverse datasets, model architectures, and self-interpretable GNN frameworks validate the effectiveness of EE.

We hope this work, which also serves as an evaluation study, will inspire researchers and practitioners to recognize explanation consistency as an indispensable criterion in the design and evaluation of self-interpretable GNNs. Future research efforts involve moving beyond reactive strategies (such as EE) and prioritizing the development of proactive strategies that directly address redundancy during model training, enabling highly trustworthy GNN systems.

## Acknowledgements

We thank Dr. Zeng Wang for her constructive discussions and valuable feedback. We also thank the anonymous reviewers for their comments and suggestions. This work was supported in part by National Natural Science Foundation of China (Grant No.62176043 and No.62072077).

## Impact Statement

This work aims to enhance the trustworthiness of explanations produced by self-interpretable GNNs, aligning with the goals of responsible AI. We do not foresee any direct negative societal consequences and expect that our work will support the safe deployment of GNNs in high-stakes domains such as healthcare and scientific discovery.

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

# A. Proofs

**Proposition 4.1.** *Define the crucial graph subset as the optimal GNN explanation $G_s^*$. When $K \geq |G_s^*|$, there exist $\geq 1$ valid GNN explanations that can satisfy Equation* (1).

*Proof.* Based on Lagrangian, adjusting $\beta$ actually corresponds to setting different budget values $K$ (Zhang et al., 2022). In our experiments, $K \geq |G_s^*|$ generally holds true in existing self-interpretable GNNs. By definition of MI, optimizing Equation (1) equals:

$$\max_{G_s \subseteq G, |G_s| \leq K} H(Y) - H(Y|G_s). \tag{9}$$

Since $H(Y)$ is a constant once the dataset is given, maximizing the above equals:

$$\max_{G_s \subseteq G, |G_s| \leq K} \underbrace{-H(Y|G_s)}_{\leq 0}, \tag{10}$$

where $H(Y|G_s) = 0$ means the graph subset $G_s$ contains enough information to predict the label $Y$, i.e., $G_s^* \subseteq G_s$. If $K$ is larger than the size of the optimal GNN explanation, i.e., $K \geq |G_s^*|$, we have $\sum_{n=0}^{K-|G_s^*|} \binom{|G-G_s^*|}{n}$ different explanations that can satisfy Equation (1). This completes the proof. $\qquad\square$

**Proposition 4.2.** *Assume that $G$ has multiple independent explanations $G_s^* = \{G_s^{1,*}, G_s^{2,*}, \ldots, G_s^{m,*}\}$ with pairwise disjoint edge sets. Then, no single edge is strictly necessary.*

*Proof.* An edge $e \in \mathcal{E}$ is strictly necessary if removing $e$ from $G_s$ causes $I(G_s; Y)$ to decrease, i.e., $e$ is indispensable for any valid explanation of $Y$. By definition, each explanation $G_s^{i,*}$ in the set $G_s^* = \{G_s^{1,*}, G_s^{2,*}, \ldots, G_s^{m,*}\}$ is sufficient to fully explain $Y$. Therefore:

$$I(G_s^{i,*}; Y) = I(G; Y) \quad \forall i \tag{11}$$

Suppose $e \in \mathcal{E}$ is considered strictly necessary. Then, its removal must render all valid explanations of $Y$ invalid. Formally, we have:

$$I(G_s \setminus \{e\}; Y) < I(G_s; Y) \quad \forall G_s \in G \text{ and } |G_s| \leq K \tag{12}$$

However, by the independence property of the explanations, each explanation $G_s^{i,*} = (\mathcal{V}_s^i, \mathcal{E}_s^i)$ satisfies the following condition:

$$\mathcal{E}_s^i \cap \mathcal{E}_s^j = \emptyset, \quad \forall i \neq j, \ i, j \in \{1, 2, \ldots, m\}. \tag{13}$$

Therefore, for any $G_s^{i,*}$ that contains $e$, there exist at least one $G_s^{j,*} (j \neq i)$ that does not rely on $e$. Thus, the removal of $e$ does not invalidate all explanations, and $I(G_s \setminus \{e\}; Y) = I(G_s; Y)$ still holds for at least one $G_s \in \{G_s^{1,*}, G_s^{2,*}, \cdots, G_s^{m,*}\}$. Since this reasoning holds for any edge $e \in \mathcal{E}$, we conclude that no single edge is strictly necessary when $G$ has multiple independent explanations. This completes the proof. $\qquad\square$

**Proposition 5.1.** *Let $X$ be a random variable with $\mathbb{E}[X] = a$ and $\mathrm{Var}(X) = b$, where $X \in [0, 1]$. Let $x_1, x_2$ be independent draws from $X$, and let $\bar{X}_1, \bar{X}_2$ be the sample means of two independent $n$-samples from $X$. For a positive hyperparameter $k \in (0, 1]$, we have:*

$$\mathbb{P}(|\bar{X}_1 - \bar{X}_2| < k|x_1 - x_2|) \geq 1 - \frac{2b}{nk^2|x_1 - x_2|^2}. \tag{14}$$

*Proof.* We aim to derive a lower bound for the probability $\mathbb{P}(|\bar{X}_1 - \bar{X}_2| < k|x_1 - x_2|)$. First, we rewrite the probability using the complement rule:

$$\mathbb{P}(|\bar{X}_1 - \bar{X}_2| < k|x_1 - x_2|) = 1 - \mathbb{P}(|\bar{X}_1 - \bar{X}_2| \geq k|x_1 - x_2|). \tag{15}$$

Next, we apply Chebyshev's inequality[4]. Chebyshev's inequality states that for any random variable $Z$ with finite mean $\mathbb{E}[Z]$ and variance $\text{Var}(Z)$, and any positive value $\epsilon$:

$$\mathbb{P}(|Z - \mathbb{E}[Z]| \geq \epsilon) \leq \frac{\text{Var}(Z)}{\epsilon^2}. \tag{16}$$

Now, we let $Z = \bar{X}_1 - \bar{X}_2$ and $\epsilon = k|x_1 - x_2|$. Applying Chebyshev's inequality, we get:

$$\mathbb{P}(|\bar{X}_1 - \bar{X}_2 - \mathbb{E}[\bar{X}_1 - \bar{X}_2]| \geq k|x_1 - x_2|) \leq \frac{\text{Var}(\bar{X}_1 - \bar{X}_2)}{(k|x_1 - x_2|)^2}. \tag{17}$$

Since $\mathbb{E}[\bar{X}_1] = \mathbb{E}[X] = a$ and $\mathbb{E}[\bar{X}_2] = \mathbb{E}[X] = a$, we have:

$$\mathbb{E}[\bar{X}_1 - \bar{X}_2] = \mathbb{E}[\bar{X}_1] - \mathbb{E}[\bar{X}_2] = a - a = 0. \tag{18}$$

Substituting Equation (18) into Equation (17), we obtain:

$$\mathbb{P}(|\bar{X}_1 - \bar{X}_2| \geq k|x_1 - x_2|) \leq \frac{\text{Var}(\bar{X}_1 - \bar{X}_2)}{k^2|x_1 - x_2|^2}. \tag{19}$$

Now, we calculate the variance of $\bar{X}_1 - \bar{X}_2$. Since $\bar{X}_1$ and $\bar{X}_2$ are the sample means of two independent samples of size $n$ from $X$, we have:

$$\text{Var}(\bar{X}_1) = \frac{\text{Var}(X)}{n} = \frac{b}{n}, \tag{20}$$

$$\text{Var}(\bar{X}_2) = \frac{\text{Var}(X)}{n} = \frac{b}{n}, \tag{21}$$

Because $\bar{X}_1$ and $\bar{X}_2$ are independent, the variance of their difference is the sum of their variances:

$$\text{Var}(\bar{X}_1 - \bar{X}_2) = \text{Var}(\bar{X}_1) + \text{Var}(\bar{X}_2) = \frac{b}{n} + \frac{b}{n} = \frac{2b}{n}. \tag{22}$$

Substituting Equation (22) into Equation (19), we get:

$$\mathbb{P}(|\bar{X}_1 - \bar{X}_2| \geq k|x_1 - x_2|) \leq \frac{2b}{nk^2|x_1 - x_2|^2}. \tag{23}$$

Finally, substituting Equation (23) back into Equation (15), we obtain the desired lower bound:

$$\mathbb{P}(|\bar{X}_1 - \bar{X}_2| < k|x_1 - x_2|) \geq 1 - \frac{2b}{nk^2|x_1 - x_2|^2}. \tag{24}$$

This completes the proof. $\qquad\square$

**Proposition 5.2.** *Let $X$ and $W$ be independent random variables with $X \in [0, 1]$ and $W \in [\delta, 1]$, where $0 < \delta \leq 1$. Let $\bar{X}$ be the sample mean of $n$ independent observations of $X$. If $\mathbb{E}[X] < \delta$, then we have:*

$$\mathbb{P}(\bar{X} < W) \geq 1 - \exp(-2n(\delta - \mathbb{E}[X])^2). \tag{25}$$

*Proof.* Since $W \in [\delta, 1]$, if $\bar{X} < \delta$, then it must also be true that $\bar{X} < W$. Therefore:

$$\mathbb{P}(\bar{X} < \delta) \leq \mathbb{P}(\bar{X} < W). \tag{26}$$

We can calculate $\mathbb{P}(\bar{X} < \delta)$ as follows:

$$\mathbb{P}(\bar{X} < \delta) = 1 - \mathbb{P}(\bar{X} \geq \delta). \tag{27}$$

---

[4]https://en.wikipedia.org/wiki/Chebyshev%27s_inequality

Next, we apply Hoeffding's inequality[5]. Hoeffding's inequality states that for several independent random variables $Z_1, \ldots, Z_n$ with $Z_i \in [a, b]$ for all $i$, where $-\infty < a \leq b < +\infty$, we have:

$$\mathbb{P}\left(\frac{1}{n}\sum_{i=1}^{n}(Z_i - \mathbb{E}[Z_i]) \geq t\right) \leq \exp\left(-\frac{2nt^2}{(b-a)^2}\right). \tag{28}$$

In our case, $X \in [0, 1]$, so $(b - a) = 1$. Applying Hoeffding's inequality with $t = \delta - \mathbb{E}[X]$ (since $\mathbb{E}[X] < \delta$, $t > 0$), we have:

$$\mathbb{P}(\bar{X} \geq \delta) = \mathbb{P}(\bar{X} - \mathbb{E}[X] \geq \delta - \mathbb{E}[X]) \leq \exp\left(-\frac{2n(\delta - \mathbb{E}[X])^2}{1^2}\right) = \exp(-2n(\delta - \mathbb{E}[X])^2). \tag{29}$$

Thus,

$$\mathbb{P}(\bar{X} < \delta) \geq 1 - \exp(-2n(\delta - \mathbb{E}[X])^2). \tag{30}$$

Combining Equation (30) with Equation (26), we obtain:

$$\mathbb{P}(\bar{X} < W) \geq 1 - \exp(-2n(\delta - \mathbb{E}[X])^2). \tag{31}$$

This completes the proof. □

## B. Spurious Correlations Are Not the Primary Drivers of Inconsistency

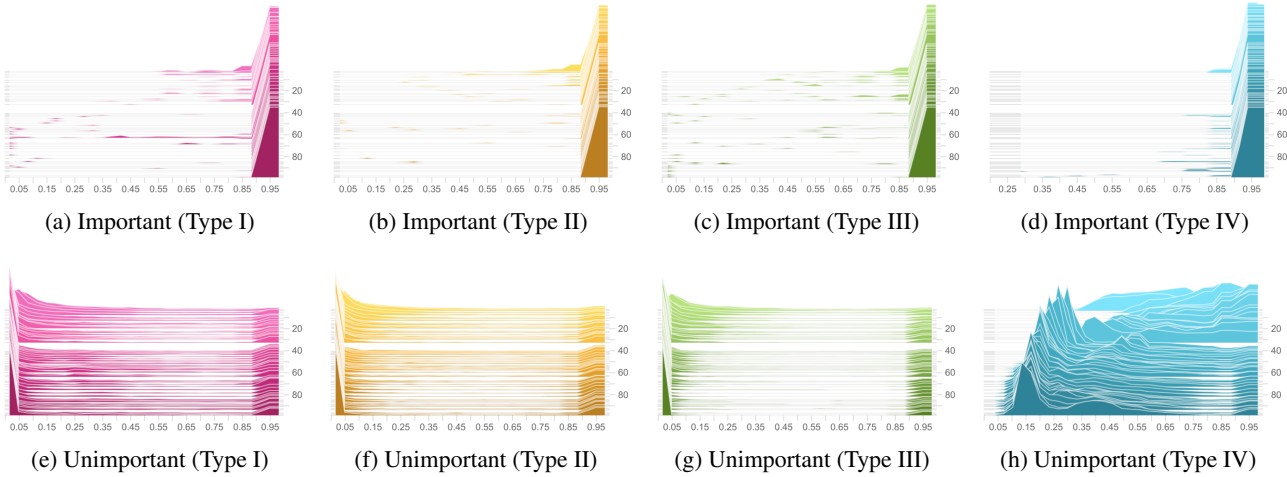

|  |  |  |  |
|---|---|---|---|
| (a) Important (Type I) | (b) Important (Type II) | (c) Important (Type III) | (d) Important (Type IV) |
| (e) Unimportant (Type I) | (f) Unimportant (Type II) | (g) Unimportant (Type III) | (h) Unimportant (Type IV) |

Figure 6: Histograms of edge weights trained on the 3MR dataset.

In this section, we present histograms of edge weights for the 3MR, BENZENE, and MUTAGENICITY datasets (see Figure 6, Figure 7, and Figure 8). The results indicate that, when evaluated on datasets without deliberately introduced spurious patterns, all self-interpretable GNNs are able to identify truly important edges. However, explanation inconsistency still persists. This suggests that spurious correlations are not the root cause of explanation inconsistency in self-interpretable GNNs. Instead, we attribute this inconsistency to redundancy: different irrelevant edges are identified as important across training runs, leading to variations in explanations.

## C. Hyperparameter Tuning Fails to Address Redundancy

To quantify the impact of hyperparameter tuning, we present histograms of $|G_s|/|G|$ for the 3MR, BENZENE, and MUTAGENICITY datasets, shown in Figure 9, Figure 10, and Figure 11, respectively. The results show that redundancy cannot be addressed by hyperparameter tuning. Increasing the value of $\beta$ generally leads to a decrease in the model's prediction accuracy. For instance, on the Type III framework, setting $\beta$ to 1 results in an accuracy reduction of 5%-20%.

---

[5] https://cs229.stanford.edu/extra-notes/hoeffding.pdf

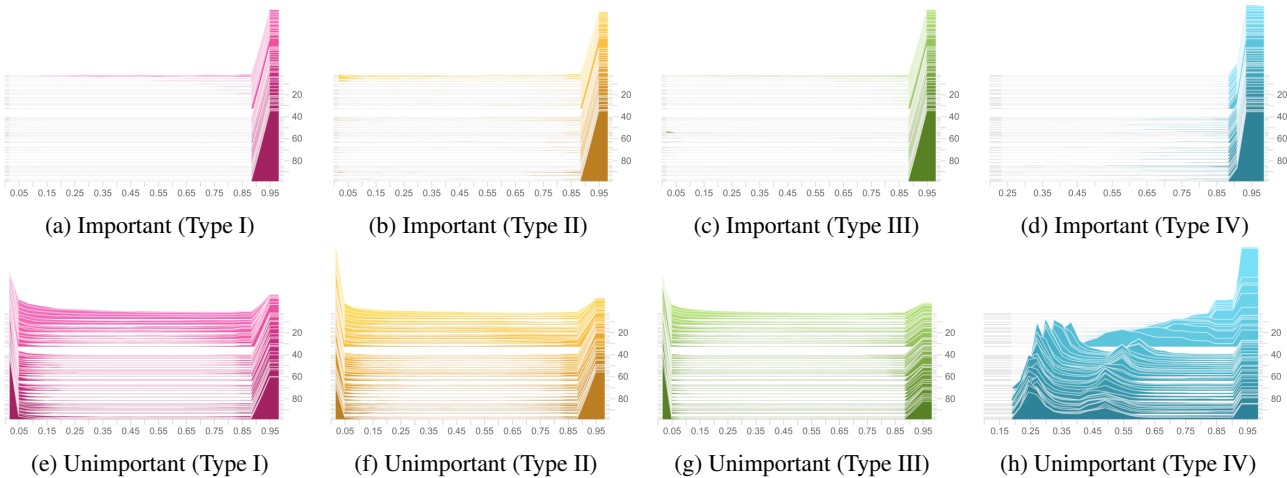

Figure 7: Histograms of edge weights trained on the BENZENE dataset.

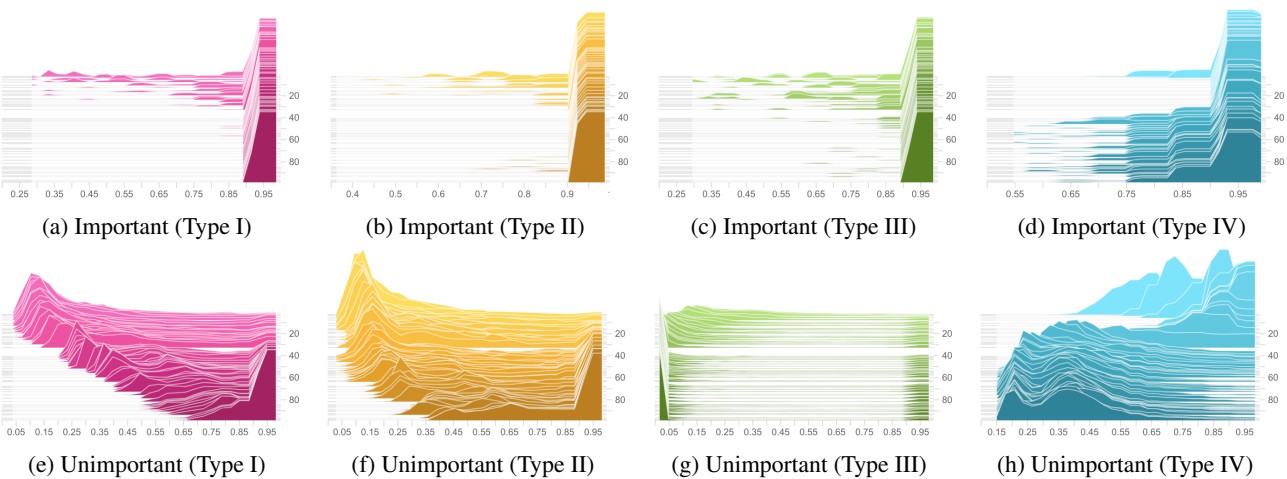

Figure 8: Histograms of edge weights trained on the MUTAGENICITY dataset.

## D. Contrastive Learning Fails to Address Redundancy

When it comes to incorporating contrastive loss into existing self-interpretable GNNs, we present our results in Table 2. "+CL (c)" means positive and negative samples were created by adding/removing $c$ edges from $G_s$. On the BA-2MOTIFS dataset, incorporating contrastive loss reduces explanation inconsistency and improves accuracy. However, on the BENZENE dataset, incorporating contrastive loss leads to performance degradation across all metrics.

## E. Rationale for Metrics and Datasets Selection

### E.1. Metrics Selection

To evaluate explanation quality, two types of metrics are commonly used in the community: those that rely on ground-truth explanations and those that do not. Ground-truth-based metrics, such as ROC-AUC, offer unbiased evaluation standards, as they directly measure how well the explanation aligns with predefined important substructures. However, their applicability is limited by the scarcity of high-quality annotated datasets. Moreover, even when such annotations exist, they may not fully reflect the internal reasoning of the model—they can rely on alternative, equally valid but human-incomprehensible patterns to make correct predictions. As a result, agreement with ground-truth does not always guarantee that an explanation truly captures the model's behavior.

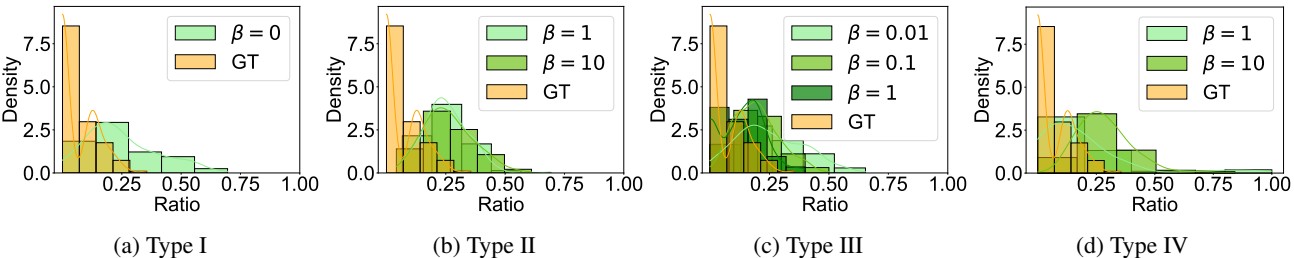

Figure 9: Histograms of $|G_s|/|G|$ on the 3MR dataset.

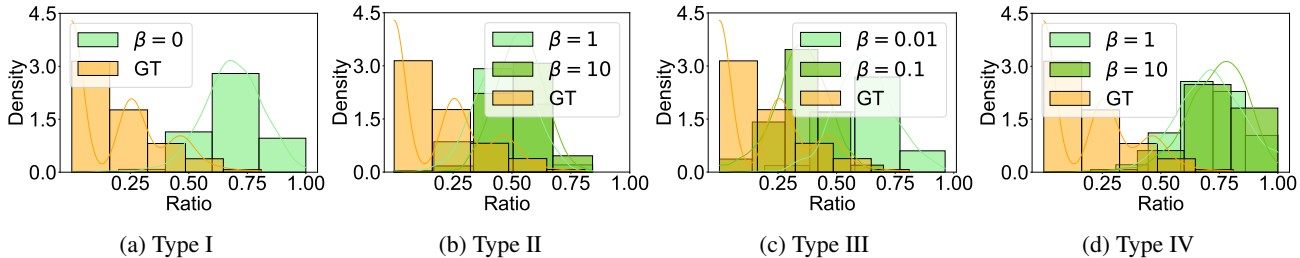

Figure 10: Histograms of $|G_s|/|G|$ on the BENZENE dataset.

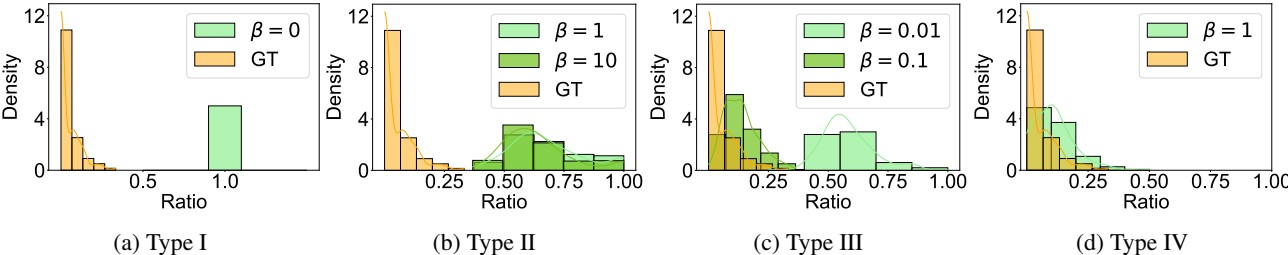

Figure 11: Histograms of $|G_s|/|G|$ on the MUTAGENICITY dataset.

In contrast, metrics that do not require ground-truth labels aim to assess *faithfulness* (Amara et al., 2022), i.e., how well the explanation aligns with the model's own decision-making process. A widely used class of such metrics is the family of *Fidelity* (FID) metrics (Amara et al., 2022), which quantify the change in the model's prediction when the identified important subgraph is either removed or retained. These metrics are more broadly applicable and, in theory, enable explanation quality to be assessed in any scenario. However, their reliability remains an active topic of debate (Faber et al., 2021), particularly due to distribution shifts introduced by input perturbations (Zheng et al., 2024). As long as OOD exists, FID-based metrics assess not only explanation quality but also the model's generalization ability (Azzolin et al., 2025), introducing confounding factors that weaken the objectivity of the evaluation.

AUC vs. FID has long been debated, and neither is perfect. The key contribution of our work is to raise awareness in the community that redundancy in explanations weakens explanation quality. Given (1) the specific nature of our task (we need ground-truth to assess redundancy), (2) that AUC's limitations can be fully addressed in certain cases, and (3) that the FID's limitations can't be fully addressed (to date), we ultimately chose AUC and just feel that SHD and AUC are enough to support our findings.

As discussed above, ground-truth explanations can sometimes be misleading, as they may not accurately reflect the reasoning process of the trained model. To avoid this risk, we were highly selective in choosing datasets and only included four well-established benchmarks that have been extensively validated by the community. Specifically, BA-2MOTIFS and MUTAGENICITY have been adopted in prior works such as GNNExplainer (Ying et al., 2019), PGExplainer (Luo et al., 2020), DIR (Wu et al., 2022b), and GSAT (Miao et al., 2022), all of which use AUC for evaluation. For 3MR and BENZENE, AUC was also employed by Rao et al. (2022). To further ensure that ground-truth explanations align with the model's

Table 2: We run each method 10 times, reporting the mean and standard deviation. Results significantly outperforming the baseline ✗ (GIN, paired t-test, $p < 0.05$) are underlined. Type I is selected for our case study.

| DATASET | STRATEGY | SHD | AUC | ACC |
|---|---|---|---|---|
| BA-2MOTIFS | ✗ | 4.75±4.68 | 99.30±0.34 | 97.10±8.70 |
| | +CL (1) | 4.27±3.90 | 99.67±0.10 | 100.00±0.00 |
| | +CL (3) | 4.77±4.38 | 99.35±0.57 | 100.00±0.00 |
| | +CL (5) | 4.71±5.36 | 99.29±0.28 | 99.69±0.90 |
| BENZENE | ✗ | 6.11±4.34 | 83.50±2.23 | 91.60±0.66 |
| | +CL (1) | 10.89±6.57 | 81.97±2.68 | 90.49±1.64 |
| | +CL (3) | 13.05±8.36 | 79.47±3.46 | 88.37±4.44 |
| | +CL (5) | 9.14±5.54 | 81.77±3.77 | 87.92±3.07 |

Table 3: Statistics of the datasets.

| | BA-2MOTIFS | 3MR | BENZENE | MUTAGENICITY |
|---|---|---|---|---|
| # Graphs | 1000 | 2877 | 12000 | 2951 |
| # Nodes / Graph | 25.00 | 20.31 | 20.58 | 30.13 |
| # Edges / Graph | 25.48 | 22.27 | 21.82 | 30.45 |
| # Node Features | 10 | 14 | 14 | 14 |
| Multiple Types of Rationales? | No | No | No | -NO2 and -NH2 |
| Multiple Identical Explanations? | No | ≥ 1 Three-membered Rings | ≥ 1 Benzene Rings | ≥ 1 Explanations |
| # Ratio of Multiple Explanations | 0.00% | 8.21% | 33.64% | 32.01% |

learned knowledge, we manually inspected the outputs of trained models. Specifically, we validated dataset reliability by checking: (1) whether, in correctly classified cases, the model assigns high importance to truly relevant edges; and (2) whether the model relies on any unintended shortcuts. While such manual inspection cannot guarantee absolute reliability, it provides a reasonable safeguard against misleading explanations.

We acknowledge that verifying the absolute faithfulness of ground-truth explanations remains a fundamental challenge. Designing more principled and robust evaluation protocols is an important direction for future work.

### E.2. Datasets Selection

We select four datasets: a synthetic dataset, BA-2MOTIFS (Luo et al., 2020), and three real-world dataset – 3MR (Rao et al., 2022), BENZENE (Morris et al., 2020), and MUTAGENICITY (Morris et al., 2020) – all sourced from the graph learning community and have ground-truth explanation labels.

**BA-2MOTIFS:** A synthetic dataset with binary graph labels created by Luo et al. (2020). Class labels are determined by house motifs and cycle motifs, which serve as the ground-truth explanations for the two classes. Each graph has only one explanation.

**3MR:** A real-world molecular property prediction dataset where nodes represent atoms and edges represent chemical bonds (Rao et al., 2022). Each graph is labeled binary to indicate the presence of one or more three-membered rings. Multiple explanations may exist for each graph.

**BENZENE:** A real-world molecular property prediction dataset where nodes represent atoms and edges represent chemical bonds (Sanchez-Lengeling et al., 2020). Each graph is labeled binary to indicate the presence of one or more benzene rings. Multiple explanations may exist for each graph.

**MUTAGENICITY:** A real-world molecular property prediction dataset where nodes represent atoms and edges represent chemical bonds (Debnath et al., 1991). Each graph is labeled binary to indicate its mutagenic effect, with -NO2 and -NH2 considered ground-truth explanations (Luo et al., 2020). Multiple explanations may exist for each graph.

These datasets are selected to represent a variety of characteristics commonly encountered in graph explanation tasks. BA-2MOTIFS, a synthetic dataset, has a single, unambiguous ground-truth explanation for each instance. 3MR, a real-world dataset, while having multiple possible explanations for each instance, has a relatively low proportion (8%) of instances with

Table 4: We run each method 10 times, reporting the mean and standard deviation. Results significantly outperforming the baseline ✗ (GIN, paired t-test, $p < 0.05$) are underlined. Values in brackets show performance changes after Explanation Ensemble. Dark green indicates performance improvement, while dark red indicates performance degeneration.

| METHOD | METRIC | STRATEGY | BA-2MOTIFS | 3MR | BENZENE | MUTAGENICITY |
|---|---|---|---|---|---|---|
| **Type I** (Attention) | SHD | ✗ | 2.74±2.75 (↓ 1.37) | 6.44±3.85 (↓ 2.91) | 0.00±0.00 (0.00) | 2.90±5.80 (↓ 2.90) |
| | | +SWA | 3.89±3.66 (↓ 2.70) | 3.06±2.15 (↓ 1.77) | 0.00±0.00 (0.00) | 7.79±5.35 (↓ 4.38) |
| | | +EA | 3.80±3.84 (↓ 2.01) | 4.82±3.07 (↓ 2.67) | 0.00±0.00 (0.00) | 0.12±0.25 (↓ 0.12) |
| | AUC (%) | ✗ | 99.12±0.17 (↑ 0.14) | 98.89±0.20 (↑ 0.56) | 83.63±4.34 (↑ 6.59) | 97.22±0.89 (↑ 1.56) |
| | | +SWA | 99.06±0.13 (↑ 0.08) | 98.56±0.12 (↑ 0.31) | 86.93±2.68 (↑ 2.38) | 98.83±0.28 (↑ 0.37) |
| | | +EA | 98.95±0.28 (↑ 0.17) | 98.78±0.15 (↑ 0.56) | 82.33±5.96 (↑ 8.30) | 96.59±1.82 (↑ 2.06) |
| | ACC (%) | ✗ | 100.00±0.00 (0.00) | 99.13±0.41 (↑ 0.17) | 91.17±0.36 (↑ 0.41) | 92.22±1.08 (↑ 0.34) |
| | | +SWA | 70.30±5.00 (↓ 2.00) | 94.15±0.58 (↓ 0.04) | 87.72±0.26 (↑ 0.36) | 89.93±0.47 (↓ 0.41) |
| | | +EA | 100.00±0.00 (0.00) | 99.37±0.33 (↑ 0.28) | 91.36±0.39 (↑ 0.14) | 92.22±0.95 (↑ 0.78) |
| **Type II** (Causal Learning) | SHD | ✗ | 1.34±1.30 (↓ 0.66) | 3.66±2.51 (↓ 2.00) | 0.00±0.00 (0.00) | 0.15±0.24 (↓ 0.15) |
| | | +SWA | 3.54±3.28 (↓ 2.20) | 4.22±2.88 (↓ 2.14) | 0.00±0.00 (0.00) | 7.15±4.82 (↓ 3.95) |
| | | +EA | 2.57±2.70 (↓ 1.76) | 4.86±3.16 (↓ 2.75) | 0.00±0.00 (0.00) | 0.01±0.03 (↓ 0.01) |
| | AUC (%) | ✗ | 97.80±0.42 (↑ 1.08) | 98.42±0.10 (↑ 0.70) | 73.66±6.03 (↑ 8.72) | 96.87±0.80 (↑ 1.59) |
| | | +SWA | 96.96±1.17 (↑ 1.67) | 97.23±0.49 (↑ 0.99) | 66.65±4.43 (↑ 8.25) | 97.83±0.43 (↑ 1.22) |
| | | +EA | 97.53±0.59 (↑ 1.33) | 98.36±0.30 (↑ 0.85) | 75.25±6.92 (↑ 7.14) | 95.27±1.51 (↑ 2.38) |
| | ACC (%) | ✗ | 78.59±4.05 (↑ 6.41) | 98.30±0.66 (↑ 1.00) | 84.20±1.73 (↑ 3.21) | 89.66±1.10 (↑ 0.88) |
| | | +SWA | 65.50±6.77 (↑ 0.50) | 88.75±1.04 (↑ 0.52) | 82.79±1.02 (↑ 2.62) | 90.03±0.48 (↑ 0.17) |
| | | +EA | 80.19±6.70 (↑ 7.81) | 99.03±0.43 (↑ 0.62) | 73.15±4.89 (↑ 3.76) | 89.72±0.83 (↑ 0.48) |
| **Type III** (Size Constraint) | SHD | ✗ | 2.77±2.57 (↓ 1.68) | 3.51±2.37 (↓ 1.62) | 0.00±0.00 (0.00) | 18.74±11.70 (↓ 9.91) |
| | | +SWA | 4.95±4.28 (↓ 3.10) | 2.89±2.22 (↓ 1.52) | 0.00±0.00 (0.00) | 11.49±6.87 (↓ 6.13) |
| | | +EA | 3.15±3.45 (↓ 1.67) | 3.53±2.41 (↓ 1.69) | 0.00±0.00 (0.00) | 11.07±8.91 (↓ 6.74) |
| | AUC (%) | ✗ | 99.11±0.16 (↑ 0.11) | 98.44±0.19 (↑ 0.72) | 85.75±4.06 (↑ 5.83) | 97.49±0.86 (↑ 1.57) |
| | | +SWA | 98.94±0.18 (↑ 0.09) | 98.37±0.09 (↑ 0.42) | 89.16±1.02 (↑ 1.73) | 98.67±0.28 (↑ 0.48) |
| | | +EA | 99.10±0.12 (↑ 0.09) | 98.35±0.23 (↑ 0.75) | 82.31±4.92 (↑ 7.82) | 97.00±1.46 (↑ 1.91) |
| | ACC (%) | ✗ | 100.00±0.00 (0.00) | 98.85±0.46 (↑ 0.45) | 90.67±0.29 (↑ 0.58) | 91.35±1.22 (↑ 0.87) |
| | | +SWA | 68.80±5.11 (↑ 0.20) | 93.04±1.08 (↑ 0.73) | 87.83±0.33 (↑ 0.25) | 89.66±0.40 (↓ 0.14) |
| | | +EA | 100.00±0.00 (0.00) | 99.30±0.40 (↑ 0.35) | 91.37±0.26 (↑ 0.13) | 91.95±1.04 (↑ 0.95) |
| **Type IV** (MI Constraint) | SHD | ✗ | 3.72±3.54 (↓ 2.25) | 5.75±2.90 (↓ 2.25) | 3.78±2.50 (↓ 2.11) | 13.53±10.86 (↓ 5.98) |
| | | +SWA | 3.13±3.58 (↓ 2.73) | 4.89±3.51 (↓ 2.86) | 0.48±0.73 (↓ 0.48) | 0.00±0.00 (0.00) |
| | | +EA | 2.33±2.34 (↓ 1.51) | 6.64±3.65 (↓ 2.49) | 4.78±2.88 (↓ 2.76) | 11.91±8.38 (↓ 5.13) |
| | AUC (%) | ✗ | 98.73±0.14 (↑ 0.20) | 99.31±0.15 (↑ 0.19) | 88.15±1.24 (↑ 1.01) | 99.15±0.19 (↑ 0.17) |
| | | +SWA | 99.21±0.16 (↑ 0.13) | 98.40±0.23 (↑ 0.24) | 89.09±0.65 (↑ 0.87) | 99.14±0.07 (↑ 0.13) |
| | | +EA | 98.16±0.33 (↑ 0.29) | 99.14±0.24 (↑ 0.24) | 86.69±1.18 (↑ 1.14) | 99.19±0.22 (↑ 0.21) |
| | ACC (%) | ✗ | 100.00±0.00 (0.00) | 99.51±0.16 (↓ 0.21) | 88.83±0.46 (↑ 0.35) | 91.58±0.57 (↑ 0.64) |
| | | +SWA | 87.30±8.03 (↑ 6.70) | 93.07±0.99 (↑ 0.27) | 86.96±0.68 (↑ 0.79) | 89.86±0.72 (↑ 0.68) |
| | | +EA | 100.00±0.00 (0.00) | 99.27±0.24 (↑ 0.38) | 87.62±0.47 (↑ 0.46) | 91.85±1.22 (↑ 0.71) |

multiple explanations. BENZENE, in contrast, exhibits a higher degree of multi-explanations, with 33% of its instances having multiple valid explanations. Finally, MUTAGENICITY also features instances with multiple explanations, and these explanations can be of varying types. This diverse selection of datasets ensures a comprehensive evaluation of our proposed method. Statistics of the datasets are provided in Table 3.

# F. More Quantitative Results

In this section, we present the results of using GCN as the model architecture (see Table 4). Some self-interpretable GNNs achieve zero SHD (indicating no inconsistency) on certain datasets, but this result is misleading. It does not imply that they produce meaningful explanations. Instead, the zero SHD is often an artifact of the model assigning weights above 0.5 to all edges. In our experiments, we used a 3-layer GCN and a 2-layer GIN, as a 2-layer GCN failed to achieve satisfactory classification performance. However, we observe that even with improved accuracy, the explanations produced by GCN remain much less meaningful compared to GIN. This may be because GCN's aggregation mechanism (i.e., neighborhood averaging) tends to dilute node-specific signals, especially as the number of layers increases. As a result, the model may become less sensitive to specific substructures and assign uniformly high attention to all edges, leading to deceptively low SHD scores. In contrast, GIN, which is more expressive and theoretically as powerful as the Weisfeiler-Lehman test, appears

Table 5: Datasets and their URLs

| DATASET | URL |
| --- | --- |
| BA-2MOTIFS | https://github.com/Graph-COM/GSAT?tab=readme-ov-file |
| 3MR | https://drive.google.com/drive/folders/1b0MowzK4LSlkih3ie1bnj6IkjRqPAKH5 |
| BENZENE | https://chrsmrrs.github.io/datasets/docs/datasets |
| MUTAGENICITY | https://github.com/flyingdoog/PGExplainer/tree/master/dataset |

to better preserve local discriminative structures and yields more valuable explanations.

## G. Experimental Settings

Considering that some of the datasets we selected were not used in the original papers of certain methods, and to ensure a (relatively) fair comparison, we built our backbone entirely upon the official GSAT (Type IV) implementation and then reimplemented ATT (Type I), GISST (Type III), and CAL (Type III) on top of this backbone. Specifically:

- ATT (Velickovic et al., 2018): This only required modifying the loss function (classification loss).

- GISST (Lin et al., 2020): This only required modifying the loss function (classification loss + L1 loss + entropy loss). In practice, we observed that due to the use of Gumbel sampling during training, the model naturally tends to produce near-binary outputs. As a result, the entropy loss became unnecessary and was ultimately omitted in our final implementation.

- CAL (Sui et al., 2022): We reimplemented CAL within the GSAT framework based on its original implementation. Besides modifying the loss function, CAL requires three classifiers, each taking different inputs, as described in Equation (3). In the original GSAT (Miao et al., 2022), the final classifier is a single linear layer. However, during our experiments, we found that 1-layer classifier(s) led to convergence issues for CAL. Therefore, we changed it to 3-layer classifier(s). For consistency, we also applied this modification to ATT, GISST, and GSAT.

**SHD Calculation.** We run each method 10 times with different random seeds. To evaluate stability, we randomly divide the 10 models into two disjoint groups and compute the SHD between their corresponding EE-generated explanations. For Table 1 and Table 4, we consider all possible 5-vs-5 splits and report the average SHD across all such pairs. For Figure 5, we vary the ensemble size $n$ from 1 to 5 and sample all possible disjoint $n$-vs-$n$ pairs from the 10 models, reporting the average SHD for each $n$.

To compute the standard deviation of SHD, we take a sample-level approach: for each individual graph, we calculate the SHD scores across all valid disjoint model groupings, then compute the standard deviation of these scores for that graph. The final reported value is the average of these per-sample standard deviations across the dataset. This reflects sample-level variability, rather than population-level variance over the whole dataset.

**Model Architectures.** GIN consists of 2 layers with a hidden size of 64, while GCN has 3 layers with the same hidden size. Following Miao et al. (2022), we employ a 3-layer Multi-Layer Perceptron (MLP) to predict edge weights, with hidden sizes set to $256, 64, 1$. Model hyperparameters are selected based on validation set performance: the learning rate is chosen from $\{0.01, 0.005, 0.001, 0.0005, 0.0001\}$ to maximize classification accuracy, and for SWA, we use either the optimal learning rate or half of it, whichever yields better validation accuracy. The coefficient for EA is selected from $\{0.01, 0.1, 1, 10, 100\}$ based on classification performance on the validation set. We start using SWA from the 10-th epoch, and at the end of every optimization step, a snapshot of the weights will be added to the SWA running average.

**Self-Interpretable GNN frameworks.** The workflow of the self-interpretable GNNs is described as follows (we use GIN as a study case). First, a 2-layer GIN is used to update node representations:

$$\mathbf{H} = \text{GIN}(G). \tag{32}$$

Next, a 3-layer MLP (explainer) with hidden size $\{256, 64, 1\}$ is used to predict edge weights. For a certain edge $(i, j)$, its edge weight is calculated as:

$$w_{ij} = \sigma(\text{MLP}([\mathbf{h}_i; \mathbf{h}_j])), \tag{33}$$

where $\sigma$ is the sigmoid function, $\mathbf{h}_i$ and $\mathbf{h}_j$ represent the representation of the $i$-th node and $j$-th node in $\mathbf{H}$, respectively, and $\sigma$ is the sigmoid function. Then, $G_s$ is factorized as:

$$\mathbb{P}(G_s) = \prod_{i,j \in \mathcal{E}} \mathbb{P}(w_{ij}), \tag{34}$$

where $\mathcal{E}$ is the set of edges in the raw graph $G$, and $\mathbb{P}(w_{ij})$ is the Bernoulli distribution $\mathrm{Bern}(w_{ij})$, representing the probability of edge $(i, j)$ being included in $G_s$. Following existing works (Miao et al., 2022; Deng & Shen, 2024; Luo et al., 2024), we employ the Gumbel Softmax technique (Jang et al., 2017) for generating GNN explanations during training. The edge weight $e_{ij}$ is calculated by:

$$\epsilon \sim \mathrm{Uniform}(0, 1), \qquad e_{ij} = \sigma((\log \epsilon - \log(1 - \epsilon) + w_{ij})/\tau) \tag{35}$$

where $\sigma$ is the sigmoid function, and $\tau$ is the temperature parameter. Finally, $G_s$ is fed to the above GIN:

$$\mathbf{z}_s = \mathrm{POOL}(\mathrm{GIN}(G_s)), \tag{36}$$

where POOL is a pooling operation $\mathbb{R}^{|\mathcal{V}| \times d} \mapsto \mathbb{R}^d$ (e.g., sum, mean, max). The other 3-layer MLP with hidden size $\{64, 64, 1\}$ is used to make final predictions:

$$y = \mathrm{MLP}(\mathbf{z}_s). \tag{37}$$

The optimization objectives defined in Section 2.2 are used during training. The hyperparameters (e.g., $\beta$, $\gamma$) are selected from $\{0.01, 0.05, 0.1, 0.5, 1, 5, 10, 50, 100\}$, primarily based on classification accuracy on the validation set. In practice, we also heuristically consider explanation conciseness during selection, and discard configurations that yield uninformative explanations. Detailed hyperparameter settings can be found in our code.

**Computing resources.** All experiments were conducted using PyTorch, trained with the Adam optimizer (Kingma & Ba, 2015), and executed on one NVIDIA RTX 4090 GPU with Intel Core i7-13700KF CPU. Running all our experiments on a single GPU took approximately 3 days.

**Open Access to Data and Code.** All datasets are published and can be downloaded from the Internet (see Table 5).

## H. Limitations

While EE is effective in improving explanation consistency and its associated accuracy through ensemble averaging, it is inherently incompatible with commonly used faithfulness metrics. These metrics typically rely on perturbing or masking parts of the input based on a single model's explanation and observing the change in its prediction (Amara et al., 2022). However, EE produces explanations by aggregating multiple models' outputs, making it unclear which model should be evaluated under perturbation. Nonetheless, EE is just a first step toward more trustworthy explanations. We envision that future solutions may achieve similar or even better results without relying on ensembling.

## I. Other Related Work

As suggested by one of the reviewers, we reviewed a set of recent works on self-interpretable methods in NLP. While these studies primarily focus on rationalization in language models, one of the papers—Multi-Generator Based Rationalization (MGR) (Liu et al., 2023)—shares certain similarities with our work. Specifically, MGR introduces multiple independently trained rationale generators to produce diverse explanations, which are then used to train a more robust predictor. Both MGR and our proposed EE adopt ensembling strategies, but they address different issues and operate in different ways: MGR targets spurious correlations and degeneration in text-based tasks and applies ensembling only during training, whereas EE, developed in the context of self-interpretable GNNs, aggregates explanations at inference time to mitigate the negative impact of redundancy. We plan to investigate whether redundancy also exists in self-interpretable language models.

