# OpenReview forum: "Redundancy Undermines the Trustworthiness of Self-Interpretable GNNs"
_ICML.cc/2025/Conference — ICML 2025 poster_

### Official Review · Reviewer_CmNK · 2025-03-11

**Overall Recommendation:** 3

**Summary:**

The paper tackles the fundamental challenge of verifying whether explanations extracted by self-interpretable models, which are considered to be more trustworthy by design, are so.
The authors highlight an issue with current approaches by providing the intuitive example (Fig. 1) of how simply changing the random seed of the model results in quite inconsistent explanations, raising concerns about their utility for model understanding. The authors identify redundancy as the root cause of this problem and test three different mitigation strategies. Among the three, only one achieves consistently good results, which involves an ensemble of explanations. The rationale of this idea is that averaging multiple explanations from diverse models can filter out noise/spurious correlation, yielding better explanations.

**Claims And Evidence:**

The claims of the paper are generally fairly well supported by arguments and experiments, except for the following cases:

- While I agree that the analysis on the redundancy and inconsistency issue of explanations is novel, the first sentence in the abstract sounds a bit of an overstatement. In fact, this is not the first work studying the trustworthiness of explanations extracted by self-interpretable GNNs [1,2,3]. I have a similar concern in line 74 of the Introduction.

- In the Type II paragraph of Section 2.2, the authors say that Causal Learning for interpretable GNNs aims at identifying
the essential parts that are responsible for the classification while ensuring that the remaining parts have no influence on the model’s output. This is, however, a shared design principle of self-interpretable models which all try to highlight the relevant portions for predictions. I would rather stress the fact that Causality-based approaches aim at learning truly causal patterns that go beyond pure spuriosity-prone statistical correlation.

- The authors claim that redundancy is somewhat unavoidable. However, they seem to be testing only a limited set of mitigation strategies, and they do not provide strong theoretical results to support this statement, making this claim not well supported in my opinion.

**Essential References Not Discussed:**

In line 35 of the Introduction, the authors ask whether Self-interpretable GNNs "truly live up to expectations?". While this is an interesting question, it should be noted that this work is not the first to pose this problem, as investigated in [1].

List of missing key references:

[1] How Faithful are Self-Explainable GNNs?

[2] How Interpretable Are Interpretable Graph Neural Networks?

[3] Reconsidering Faithfulness in Regular, Self-Explainable and Domain Invariant GNNs

[4] XAI and Bias of Deep Graph Networks

[5] Towards Robust Fidelity for Evaluating Explainability of Graph Neural Networks

**Experimental Designs Or Analyses:**

In "Run 1" of Figure 1, the authors point out that the triangle is erroneously highlighted as important by the model, and they claim that this proves the inconsistency issue outlined in the paper. However, it remains unclear whether such a three-membered ring is indeed not correlated with the label and whether it is in fact a portion of the input that the model is *not* using. In fact, if the dataset presents some biases, such a triangle might be spuriously correlated with the label, and the model might be *authorized* to exploit it, resulting in the explanation for "Run 1" as a faithful explanation for a model affected by spurious correlation.

Other than that, I found the experimental results hard to follow and to contextualize, in the sense that it is not clear which is the model under investigation. In Tables and Figures, in fact, authors refer to the tested models as Type I/II/III/IV. However, it is not clear which is the underlying model implementation being in use (GSAT vs DIR vs ...). It would be beneficial to refer to the individual architectures with their name to improve clarity.

**Methods And Evaluation Criteria:**

Authors claim that "datasets with ground-truth explanations provide a reliable benchmark for comparison". However, ground truth explanations are reportedly found to be potentially misleading, as pointed out in (Faber et al. 2021) and [4]. In general, when comparing against an explanation ground truth, it is required to ensure that the model actually learned the expected ground truth, but this issue seems not to be addressed in the paper.

Perhaps more importantly, the paper does not provide any analysis on the faithfulness of the resulting explanation, constituting a major lack in the experimental investigation of the trustworthiness of explanations.

**Other Comments Or Suggestions:**

- The caption of Figure 4 should be self-contained: (i) Which is the model under investigation? (ii) Is the loss referred to train or validation split? (iii) Is the entire loss plotted or only the explanation regularization loss? If the entire loss is plotted, are you sure that the instability is caused by the difficulty in differentiating positive vs negative samples and not by BENZENE being more difficult to fit?

- $h_{G_{s}}$ is not defined in Equation 1

**Other Strengths And Weaknesses:**

The paper is well-written and clear. Also, figures are curated and well presented, albeit captions are often not self-explanatory.

The mitigations implemented in this work are relatively new in the context of GNN explanations, but they seem not to bring solid empirical evidence, leaving open the question of whether redundancy is really unavoidable. On the same vein, the proposed solution of averaging explanations from multiple models is very easy, and albeit working well in practice, little is discussed about the limitations in terms of faithfulness of the resulting explanation and the utility of this ensable in debugging model behaviour. Therefore, to me, the practical benefits of the proposed solution for an end user remain unclear.

**Questions For Authors:**

1. How does the Explanation Ensamble solution affect the computation of other metrics for explanation quality, such as Faithfulness/Fidelity? My feeling is that since the explanation is aggregated over different models, it is no longer possible to evaluate how faithful the explanation is wrt a single model.

2. Regarding the experiments with known ground truth, do the authors check that the ground truth is actually the knowledge encoded in the model being explained and that no shortcuts have been learned? Can you please argue about the impact that this may have on your evaluation?

3. Regarding Run 1 in Figure 1, can you please argue or provide evidence regarding the fact that the model is indeed not using the triangle to predict the class? For example, an argument would be to show that such a three-membered ring is in fact present also in the other class.

I updated my score from 1 to 3 after the rebuttal by the authors that addressed most of my concerns.

**Relation To Broader Scientific Literature:**

To the best of my knowledge, there is no prior work studying explanations of self-interpretable GNNs under this lens. However, the paper fails in contextualizing it with some recent findings in the literature of analyzing the trustworthiness of explanations of those models, like [1,2,3,5].

**Theoretical Claims:**

I did not find major issues in the theoretical claims. Below are some on-point comments:

- Equation 1 formalizes only self-interpretable models with explicit size constraints (<=K), but there exist models that optimize for other objectives, like GSAT (Miao et al., 2022). In fact, in Section 2.2 the authors correctly distinguish between different training methodologies, making Equation 1 not very precise.

- In the statement of Prop. 1, the meaning of "crucial subgraph" and "valid explanations" should be clarified. Albeit intuitive, the formality of the statement would benefit.

- Similarly to above, the notion of "strictly necessary" is not defined.

---

> ### Author Rebuttal · Authors · 2025-03-27
>
> Thank you for reviewing our paper. Below we address your concerns.
>
> (1) Overstatement of the First Systematic Investigation
>
> We have addressed this in our response to Reviewer 6xUG, where we acknowledge the mistake and clarify our intent.
>
> (2) About Evaluation Metrics
>
> Faber et al. (2021) argue against removing-based evaluation and support ground-truth explanations. They suggest using datasets with reliable ground truth (please see Sec 2.2 and Sec 7 in their paper). **We acknowledge that ground-truth explanations can sometimes be misleading, so we were highly selective in choosing datasets and only included four well-established benchmarks that have been extensively validated in the community:** BA-2MOTIFS and MUTAGENICITY have been used in GNNExplainer, PGExplainer, DIR, and GSAT, and they use AUC for evaluation. For 3MR and BENZENE, Rao et al. (2022) -- the creators of the 3MR dataset -- also use AUC. We also manually inspected models' outputs to ensure that the ground-truth explanations reflect the knowledge encoded in model and that no shortcuts is learned (some works questioned the validity of the MUTAGENICITY, e.g., Both NO2 and NH2 are important, but explainer may detect only one of them, we have verified that such issues never occur in our experiments).
>
> **EE is indeed not compatible with FID, but this was not the reason we opted against FID. EE is just a first step, and future solutions may not require ensembling.** While RFID mitigates OOD, it does not fully eliminate it and compromises evaluation effectiveness (i.e., no OOD = invalid evaluation). As long as OOD exists, FID-based metrics assess not only explanation quality but also the model’s generalization ability [3], introducing confounding factors that weaken the objectivity of the evaluation. AUC vs. FID has long been debated, and neither is perfect. The key contribution of our work is to raise awareness in the community that redundancy in explanations weakens explanation quality. Given (1) the specific nature of our task (we need ground-truth to assess redundancy -- Figures 2, 3), (2) that AUC’s limitations can be fully addressed in certain cases [5], and (3) that the FID's limitations can't be fully addressed (to date), we ultimately chose AUC and just feel that SHD and AUC are enough to support our findings.
>
> PS: Beyond OOD, redundancy also weakens the validity of FID-based metrics -- in the most redundant case (i.e., $G_s = G$), FID metrics achieve trivially optimal scores (see Sec. 5.1 in [5]). Some alternatives, such as H-FID in GStarX and advanced Nec in [3], could potentially be used to evaluate redundancy but require careful design. We prepare to include a more detailed discussion in the revised version.
>
> (3) Unavoidable Redundancy
>
> We understand your concern. Our work investigates all relevant techniques to date and, through a combination of theoretical and empirical analysis, demonstrates that redundancy is challenging to eliminate (via existing techniques). Specifically: (1) Figure 3 and Prop. 4.1 illustrate why tuning hyperparameters cannot eliminate redundancy; (2) Figure 4, Prop. 4.2, and Table 2 show the limitations of the +CL. We will revise the subsection title to “On the Difficulty of Eliminating Redundancy” and hope this addresses your concern.
>
> (4) Responses to Other Concerns
>
> - Causal Learning for Interpretable GNNs: Thank you for your comment. We will revise it accordingly.
> - Equation 1: The goal of GNN explanation is to identify subgraphs that are both informative and concise. You are correct that some methods optimize for different objectives, but some papers define the GNN explanation task more generally (see Definition 2 in [5]). Even GSAT, though it constrains subgraphs from an MI perspective, still imposes a size constraint in a broader sense (see discussion in https://github.com/Graph-COM/GSAT/issues/2). We are open to revising the paper based on your suggestion.
> - Definition in Prop. 1: The crucial subgraph is the ground-truth explanation $G_s^*$, and a valid explanation must include $G_s^*$ while also satisfying the size constraint (i.e., $G_s^* \subseteq G_s$ and $|G_s| \leq K$).
> - Definition of strictly necessary: An edge is strictly necessary if its removal alone significantly impacts the prediction.
> - Model Types and Figure 4: Type I is Attention, Type II is GISST, Type III is CAL, and Type IV is GSAT. The underlying backbone is implemented using GSAT, with slight modifications to accommodate others. In Figure 4, we use Type I for evaluation (other types yield similar results, which we can provide in the revised version if needed). We plot the contrastive loss during training.
> - $h_{G_s}$ generates $G_s$, uses it as input, and outputs a graph representation.
> - Run 1 in Figure 1: The three-membered ring is also present in the other class.
>
> We appreciate your time and feedback, and we hope our rebuttal addresses your concerns. We look forward to any further discussions if needed. Thank you!

---

> > ### Comment · Reviewer_CmNK · 2025-04-03
> >
> > thank the reviewers for addressing my concerns. Please find below some follow-ups.
> >
> > **(2) About Evaluation Metrics**
> >
> > I agree that evaluating explanations wrt a ground truth is not wrong per sé. Still, it requires the experimental setting to be *carefully* designed to ensure that no alternative explanation potentially invalidating the expected ground truth exists. I feel aligned with the authors on this, and I appreciated the response. However, I would like further clarification on the following points:
> >
> > > We only included four well-established benchmarks that have been extensively validated in the community
> >
> > I'm still doubtful that current *well-established benchmarks* are actually really suited for ground truth-based evaluation. For example, see [4] Table 2, highlighting that BA-2MOTIFS admits an alternative classification rule achieving perfect classification based on average node degree rather than on finding the expected motifs.
> >
> > I don't want to overly penalize the authors on this issue, as I acknowledge that this is rarely discussed in previous papers, and it is, anyways, an issue not introduced in the authors' contribution. I believe, however, that it is important for the community to discuss this issue clearly to advance the field.
> >
> >
> > **AUC vs FID**
> >
> >
> > > EE is indeed not compatible with FID, but this was not the reason we opted against FID.
> >
> > After the clarification of the authors, I understand better the scope of the contribution, which is inherently focused only on explanation accuracy, and this justifies their focus on AUC. Nonetheless, I'm still concerned about the proposed solution. Even if we were given a golden FID metric (say, FID*), which does not suffer from OOD issues and estimates perfectly the faithfulness of the explanation without any confounding, how would EE relate to FID*? I fear that EE would still be unsuitable for FID* as it involves averaging across multiple models. While this is fine if the focus of the analysis is only on AUC, the paper would benefit from a more detailed analysis of this in the Limitations section.
> >
> > In this sense, I feel that this part of the paper can be slightly misleading; given that the authors cannot evaluate FID, they argue that FID is unsuitable for self-interpretable GNNs. However, to the best of my knowledge, FID-like metrics, despite their issues, are the current state of the art in evaluating the faithfulness of an explanation, and no strong evidence is provided to prove that FID is meaningless. I believe that the paper would benefit from a more objective discussion on the trade-off between EE and faithfulness evaluation.
> >
> >
> > **(4) Responses to Other Concerns**
> >
> > > The underlying backbone is implemented using GSAT, with slight modifications to accommodate others.
> >
> > Could you please provide more details on the implementation side?
> >
> > I do not understand whether Type II and Type III models are implemented as described in the original GiSST and CAL paper or if the authors proposed a different implementation, enriching the GSAT architecture with the regularization losses from GiSST and CAL. If that applies, to favor reproducibility, it should be clearly stated in the details of the paper that authors do not use the original implementation.
> >
> > Considering the answer, and conditional on applying the changes that I highlighted in my current response, I'm open to increasing my score from 1 to 3.

---

> > > ### Author Response · Authors · 2025-04-03
> > >
> > > We sincerely appreciate your willingness to reconsider our paper and potentially increase your score!!!
> > >
> > > **(1) On the Reliability of Datasets**
> > >
> > > We sincerely appreciate your understanding on this matter. We validated the dataset reliability by visualizing the model’s output: (1) when classification is correct, whether the model assigns high weights to truly important edges; (2) whether the model exploits any unintended shortcuts. To be honest, we acknowledge that validating the absolute reliability of a dataset is infeasible. Therefore, in the revised version, we will include a detailed description of our dataset validation process and discuss the potential issue about this.
> > >
> > > **(2) AUC and FID**
> > >
> > > We fully agree that evaluating faithfulness is crucial, and we will add a dedicated "Discussions" section in the revised version to provide an in-depth analysis of AUC and FID.
> > >
> > > PS: We completely agree with your point -- no matter how perfect FID* is, EE would still be unsuitable for it. That said, we just want to be fully transparent about our thought process during the rebuttal stage. To be honest, we did not deeply consider the relationship between EE and FID when writing the paper. We will provide a more detailed discussion of EE's limitations (computational complexity and its incompatibility with FID) in the "Limitations" section.
> > >
> > > **(3) Implementation Details**
> > >
> > > Considering that some of the datasets we selected were not used in the original papers of certain methods, and to ensure a (relatively) fair comparison, we built our backbone entirely upon the official GSAT implementation and then reimplemented ATT, GISST, and CAL on top of this backbone. Specifically:
> > > - ATT: This only required modifying the loss function (classification loss).
> > > - GISST: This only required modifying the loss function (classification loss + L1 loss + entropy loss).
> > > - CAL: We reimplemented CAL within the GSAT framework based on its original implementation. Besides modifying the loss function, CAL requires three classifiers, each taking different inputs ($G_s$, $\bar{G}_s$, and $G_s \cup \bar{G}_s$), as described in Eq. (3). In the original GSAT, the final classifier is a single linear layer. However, during our experiments, we found that one-layer classifier(s) led to convergence issues for CAL. Therefore, we changed it to three-layer classifier(s). For consistency, we also applied this modification to ATT, GISST, and GSAT.
> > >
> > > We will provide a more detailed description of the implementation in the revised version and release our code to ensure full reproducibility.
> > >
> > > If you have any further suggestions, we would greatly appreciate it if you could update your original review so that we can see them. We assure you that we will revise the paper according to your suggestions. Once again, we sincerely appreciate your constructive feedback and the opportunity to improve our work. Thank you :)

---

### Official Review · Reviewer_DDzK · 2025-03-12

**Overall Recommendation:** 3

**Summary:**

The paper aims to systematically investigate trustworthiness of explanations provided by self-Interpretable GNNs. That is, GNNs that simultaneously act as classifiers and explainers. Such GNNs highlight a subgraph as an explanation for a given graph. They provide a brief taxonomy of different  self-Interpretable GNNs. They identify the fact that different GNNs provide inconsistent explanations and not completely trustworthy explanations.
For inconsistency, they identify two reasons: (1) training instability and (2) spurious correlations. They show that conventional methods to overcome training instability do not consistently work for GNNs. While they argue that self-interpretable GNNs are better protected against spurious correlations.
They also identify redundancy as a key factor for inconsistent explanations in self-interpretable GNNs.
They show that the recently introduced criterions for necessity and sufficiency are often not simultaneously achievable.

Finally, they propose an ensemble based aggregation method to get explanations. And formally show that these explanations are more consistent with high probability compared to model level consistency.

**Claims And Evidence:**

Claim 1: GNN extracted explanations not trustworthy due to inconsistencies emerging from:
- C1.1 Training instability
- C1.2 Spurious correlations:
- C1.3 Redundancy

Evidence: The paper provides strong empirical evidence for C1.1 and C1.3. About claim C1.3 --- I agree with the larger message that spurious correlations can lead to misleading explanations.
But I do not see why authors claim "In contrast, self-interpretable GNNs simultaneously learn explanations and predictions, naturally embedding explanations into the model’s decision-making process and thus more robust to spurious features." I think if the point is that Self-interpretable explanations are more faithful to how the model works, then this statement is true. However, I believe SE-GNNs are not less likely to learn a model (and hence explanations) based on spurious correlations.

Claim 2: Ensemble of explanations  lead to lesser inconsistency than normal explanations.

Evidence: The authors show this result theoretically by bounding the inconsistence in the ensemble explanations wrt the conventional explanations (Prop 5.1). And they also show that with high probability relevant edges will be distinguished from irrelevant edges.

I do not think that the proof/claim are wrong. But I am not sure the results really reflect the true picture, my main concern is that the amount of samples required to make sure that Prop 5.1 and Prop 5.2 on graph level (i.e. for all edges w.h.p) will be quite high (one will additionally need a union bound to show this), and this analysis should be done and presented.

**Essential References Not Discussed:**

I think all the reasonably relevant paper are cited.

**Experimental Designs Or Analyses:**

- The paper empirically evaluates the inconsistency between explanations over different random seeds. And compares consistency of  ensemble based explanations to other explanation methods using Structural Hamming Distance (SHD)

- They also compare their approach against other methods wrt accuracy of the explanation.

**Methods And Evaluation Criteria:**

The paper is a relatively easy read. Theoretical claims are clearly presented and experimental results are well-explained.

**Other Comments Or Suggestions:**

-- Implications of Prop 5.1 and Prop 5.2 should be given a more nuanced analysis
-- The formal notion of explanations captured by EE should be discussed, if not completely formalized. I think having a better understanding of what these explanations represent is fundamental to the quality of this paper.

**Other Strengths And Weaknesses:**

Strengths:
- The paper is quite clearly written and makes some pertinent observations regarding self-interpretable GNNs.

Weakness:

- I see two main weakness:
(1) (Minor) The theoretical results (Prop 5.1 and Prop 5.2) are quite remote from actually supporting  the experiments. I think the number of samples required wrt these results would not justify your experimental observation of requiring only 2 runs. I believe the real reason for superior results wrt accuracy is that you potentially capture a large sufficient explanation, by aggregating many of them.

(2) (Major) I think aggregating over explanations could potentially lead to strange semantics for explanation, so lets say one benzene ring suffices to classify a molecule into a positive, with the EE, one will highlight all possible benzene rings in a positive molecule and not just one. But stranger things may happen when you have over-lapping subgraphs as potentially different explanations, your method may highlight their union.

**Questions For Authors:**

-- I may have missed this, but I did not understand where you show that conciseness constraints are set overly relaxed in other explainability methods?

**Relation To Broader Scientific Literature:**

The paper is quite relevant to the larger GNN explainability literature.

**Theoretical Claims:**

Theoretical claims are clearly presented in the paper. Although the proofs are in the appendix the statements are clear, plausible and relatively clear to see given the larger context of the paper.

My main theoretical concern is regarding Prop 5.2 and Prop 5.1, and are explained above.

---

> ### Author Rebuttal · Authors · 2025-03-29
>
> Thank you for reviewing our paper. Below we address your concerns.
>
> (1) Robustness of Self-Interpretable GNNs to Spurious Correlations
>
> We have addressed this in our response to Reviewer 6xUG, where we acknowledge the mistake and clarify our point.
>
> (2) Graph-Level Analysis of EE (Theoretical)
>
> The graph-level inconsistency is the average of edge-level inconsistencies, which means that **we do not require each edge to independently satisfy the bound in Eq. 6. Instead, the key idea is to show that the average inconsistency decreases, which is sufficient to guarantee overall improvement.**
>
> For an edge $i$, let $A_i^n \in [0, 1]$ denote the inconsistency score under EE (with $n$ samples ensemble), and $B_i$ denote the inconsistency score of two individual models (independent of $n$). At the graph level, we aim to bound:
> $$
> \mathbb{P}\left(\frac{1}{|G_s|} \sum_{i=1}^{|G_s|} A_i^n < B \right),
> $$
> where $B = \frac{1}{|G_s|} \sum_{i=1}^{|G_s|} B_i$. Using Hoeffding’s inequality (Eq. 28), we have:
> $$
> \mathbb{P}\left(\frac{1}{|G_s|} \sum_{i=1}^{|G_s|} (A_i^n - \mathbb{E}[A_i^n]) \geq B\right) \leq \exp(-2nB^2).
> $$
> Since $\mathbb{E}[A_i^n] = 0$ (Eq. 18), we get:
> $$
> \mathbb{P}\left(\frac{1}{|G_s|} \sum_{i=1}^{|G_s|} A_i^n < B\right) \geq 1 - \exp(-2nB^2).
> $$
> Thus, we have established a lower bound on the probability that EE outperforms the vanilla version, and this bound increases with $n$.
>
> Unlike Prop. 5.1, Prop. 5.2 already reflects the graph-level behavior of EE. By setting different definitions for $X$ and $W$, Prop. 5.2 serves different purposes. For instance, we can let $X$ denote the most important irrelevant edge (i.e., the one with the highest score in irrelevant edges) and $W$ denote the least important relevant edge (i.e., the one with the lowest score in relevant edges). Under this setting, Eq. 7 essentially computes the probability that the AUC reaches 100%.
>
> (3) Why EE Works (Empirical)
>
> You raised a concern that EE might lead to “strange semantics” in explanations, such as highlighting all benzene rings instead of just one. We'd like to clarify that **our goal is to identify all label-relevant structures in the graph. If a molecule contains two benzene rings, both sets of edges should be considered relevant -- this is determined by the dataset’s ground-truth explanations.**
>
> Moreover, the concern that EE might merge overlapping subgraphs into their union is also unlikely in practice. **Same structures tend to receive similar importance scores due to their structural and attribute similarity.** This means that if multiple benzene rings exist, their scores will generally be close (see Figure 5). Similar cases can be observed in GSAT’s results (e.g., Figure 3 and Figure 10 in their paper).
>
> The key reason why EE is effective is that truly important edges consistently receive high weights, so their average remains high. In contrast, irrelevant edges exhibit variance and tend to have lower average importance after EE. In other words, EE does not simply merge explanations -- it refines them by leveraging ensemble averaging to filter out noise.
>
> (4) Evidence that Conciseness Constraints are Overly Relaxed
>
> As mentioned in Line 178, classic self-interpretable methods like DIR and GSAT recommend retaining 50%-80% of edges to achieve a desirable trade-off between training stability and explanation conciseness. Additionally, our experimental results provide strong evidence of this issue (see Figure 2 and Figure 3).
>
> We appreciate your time and feedback, and we hope our rebuttal addresses your concerns. We look forward to any further discussions if needed. Thank you!

---

### Official Review · Reviewer_6xUG · 2025-03-14

**Overall Recommendation:** 3

**Summary:**

The authors study the (lack of) reliability of GNN explanations, focusing on self-explainable architectures, which promise precisely to output more reliable explanations.  They notice that these models however produce unstable explanations, or more specifically that their explanations vary even substantially between seeds, even when achieving high task accuracy.  Then, they propose and evaluate a simple mitigation strategy based on averaging explanations obtained from multiple seeds.

**Post-rebuttal update**: the authors promised to clarify several rather central issues, which is enough for me to be weakly positive about the contribution.

**Claims And Evidence:**

Main claims:

- Claim: This is the first systematic investigation of explanation reliability of GNNs
    - Evidence: There is at least one other systematic investigation in the literature, see below.

- Claim: SI-GNN explanations are "redundant".
    - Evidence: SHD values in Table 1 + convincing examples.

- Claim: SI-GNN explanations "successfully identify key features but also overemphasize some irrelevant ones" (p 1)
    - Evidence: only empirical.  They authors measure the structural similarity between ground-truth explanations and produced explanations in Table 1 for four architectures and on four data sets.  Theoretical guarantees are not provided, and I have strong doubts that any could be given in general, see my example below.

- Claim: Redundancy is difficult to eliminate
    - Evidence: none of the strategies really help in Table 1.  It is unclear to me whether averaging constitutes a "difficult strategy" (it probably does not: it is easy to set up and use.)  Other evidence includes presumably Proposition 4.2, although I do not find the link obvious. [**Q**] Is there any link between Prop 4.2 and the claim?  Could you please explain it clearly?

- Claim: Existing techniques fail to address this challenge
    - Evidence: +SWA, +EA rows in Table 1.

- Claim: Aggregating explanations from multiple seeds helps
    - Evidence: ideally, Prop 5.1 and 5.2.  However, the link to the averaging technique makes an (I think) unfounded assumption that members of the ensemble will assign higher score to truly relevant (i.e., plausible) subgraphs, and I could not find evidence for this.

**Essential References Not Discussed:**

The quality of explanations produced by SI-GNNs was also systematically analyzed in a recent paper:

  https://openreview.net/forum?id=kiOxNsrpQy

There, the authors found that SI-GNN explanations can be "insufficient" (which is, to the best of my understanding, equivalent to what the authors call "redundancy").  So, this issue was already pointed out.  The paper also proposes several potential solutions, which may be worth mentioning.

**Experimental Designs Or Analyses:**

The experimental setup is mostly good.  Table 1 is difficult to read.  The main take-away is clear enough: averaging generally helps (see the sea of green numbers).  But the relative performance of different models and competitors is very difficult to read.

Perhaps I missed it, but I was expecting to see a table or plot showcasing the correlation between reduction in SHD (explanation stability) on one side and improvement in AUC (plausibility) on the other.  Figure 6 seems to indicate that they may be anti-correlated.  [**Q**] Could you please elaborate on this?

I also have a couple of other issues with the experiments, which I'll turn into questions.

- [**Q**] Why did you choose a threshold of $0.5$ for the relevance discretization step?  Does a single constant threshold make sense across models?

- [**Q**] Why did you not evaluate the approach of Deng & Shen empirically? Given that it is supposed to underperform, numerical evidence would have provided further support for your claims. (Note that Proposition 4.2 does not say anything specific about the performance of their method.)

**Methods And Evaluation Criteria:**

The choice of data sets, architectures and metrics is appropriate.

**Other Comments Or Suggestions:**

- The notion of "truly relevant edges" should be introduced in Sec 2 or 3, and it should be clearly linked to plausibility and/or faithfulness, for ease of understanding. (I would expect GNN researchers to be familiar with these notions.)

**Other Strengths And Weaknesses:**

#Strengths

- Clarity: the paper is mostly nicely written and well structured.
- Significance: it is generally assumed that explanations output by SE-GNNs -- an increasingly popular class of models -- are high-quality, while in practice this may not be the case; this paper tackles this very issue, and as such it is definitely significant.
- Originality: to the best of my knowledge, the issue identified by the authors is novel.
- Quality: I appreciated the experiments in 3.1 and 3.2.
- Quality: standard deviations over 10 seeds.

#Weaknesses

- Clarity: The authors use terms without describing them properly.  For instance, what is an "optimal" explanation?  What is the difference between $G_s$ and $G_s^*$?  Moreover, I find it difficult to understand when the authors refer to explanations being high-quality because they are *plausible* (they capture the variables that have a causal role in the data generation process) vs because they are *faithful* (they capture the variables that are causal for the learned model).  Is an optimal explanation faithful? plausible? sufficient?  Also, necessity is only defined in enough detail in the proof of Prop 2.
- Clarity: [**Q**] How did you determine the truly important edges in Fig 2? At this stage in the paper, this is not clear.  Also, are these edges "truly important" for the model or for the data generating process?
- Quality: The proposed solution is very simple (not an issue), but I think that it fails to really address the core issue.  On the other hand, I see it more as a starting point than as a key contribution, so I am not too concerned about this limitation.

**Questions For Authors:**

Please see my questions above, I marked them with [**Q**].

In addition:

- p 4: "In contrast, self-interpreable GNNs [...] and thus more robust to spurious features."  I tend to disagree with this statement:  I don't see why SI-GNNs trained on biased data would output explanations that contain no bias.  Imagine that I manipulate the data such that a given subgraph S (say, a star) is strongly discriminative for class 0 (i.e., has ~1.0 correlation with the ground-truth label) in the training set but not in the test set (where I can simply delete it).  In turn, subgraph S would very likely be highlighted by the model's explanations.  Simply "simultaneously learning explanations and predictions" (p 4) does not prevent SI-GNNs from picking up subgraph S as relevant.  In fact, concept-bottleneck models -- which are architecturally similar to SI-GNNs, except for image inputs -- are known to suffer from shortcut learning:

  Bahadori and David Heckerman. "Debiasing Concept-based Explanations with Causal Analysis."  ICLR.

and I don't see what would give SI-GNNs an advantage over them.  To me, this sentence seems clearly wrong and I think it should be dropped.  Moreover, I don't see how the mechanisms listed in Sec 2 could fix the issue: finding coincise explanations does not help if the confound is small.  I am very skeptical that SI-GNNs (or any GNN, really) can attain high plausibility -- with guarantees -- unless strongly nudged through supervision or architectural bias.  I would appreciate a clarification.

- The manuscript relies on $G_s$ being a subgraph of $G$, but in practice -- as readily admitted by the authors -- SI-GNNs output continuous per-edge relevance scores, which need to be somehow converted into a subgraph $G_s$.  Does the construction of the discretization step somehow affect the issue of redundancy?

I am willing to **increase my score** provided the authors clarify my doubts during the rebuttal period.

**Relation To Broader Scientific Literature:**

The related work does an overall reasonable job at positioning the paper within the context of the broader literature, with out exception, see below.

**Theoretical Claims:**

I did look at the proofs in the appendix.

- One issue is that the propositions in the appendix are misnumbered.

- Prop 1: the first two sentences of the proof of Prop 1 do not belong to the proof and should be moved elsewhere (e.g. after the proof).  Also, it seems that the proof assumes that the optimal SI-GNN explanation attains $H(Y|G_s) = 0$, which may not be the case.  If it does, then all supegraphs of $G_s$ (of size up to $K$) will also have zero conditional entropy, but if it does not, they may have *lower* entropy, and therefore $G_s$ is not even an optimum.  [**Q**] What happens in this case?  Why doesn't this assumption appear in the statement of the proposition?  I am confused by what is meant by $G_s$ and what properties it should have.  What does it mean that it is "optimal"? According to what criterion?  I also skimmed through (Zhang et al., 2022), but I couldn't find the result you are referencing in p 12.  [**Q**] Could you please provide more precise coordinates?

- Prop 2: the definitions of sufficiency and necessity used here differ from others in the literature (see the openreview link below), but are intuitively sensible.  The claim is otherwise quite intuitive.

- Prop 3 and 4: appear to be correct.

---

> ### Author Rebuttal · Authors · 2025-03-31
>
> Thank you for reviewing our paper. Below we address your concerns.
>
> **Constructive Feedback We Appreciate and Will Revise:**
>
> (1) Overstatement of the First Systematic Investigation
>
> Our intent was to emphasize that we are the first to investigate the inconsistency and its associated inaccuracy -- two key aspects of trustworthiness -- in SI-GNNs. We acknowledge that our current wording has overstated this claim, and we will revise our description accordingly.
>
> (2) Robustness of Self-Interpretable GNNs to Spurious Correlations
>
> Our intent was to highlight that the inconsistency observed in post-hoc GNN explanations, which has been attributed to spurious correlations, does not apply to SI-GNNs. Yes, you are absolutely right. We appreciate your insightful comment and will revise our description accordingly.
>
> **Points We Would Like to Clarify:**
>
> Based on your comments, we realized that clarifying the notions of $G_s$, $G_s^*$, optimal explanations, and truly important edges is crucial for you to reassess our work. $G_s^*$ is the ground-truth explanation (also referred to as the optimal explanation) available in our datasets. $G_s$ is the explanation generated by SI-GNNs. Edges that belong to $G_s^*$ are what we refer to as truly important edges. Since the definition of "plausibility" varies slightly across different papers, if it is understood as "how closely $G_s$ matches $G_s^*$", then it is correct to say that we aim for $G_s$ to be plausible.
>
> (1) Clarification on Related Concerns
>
> - Evidence for Redundancy, Identification of Truly Important Edges in Figure 2, and Support for Prop. 5.2: Because $G_s^*$ is available in our datasets, we can directly evaluate whether the model assigns high weights to these truly important edges. Fig. 2(a–d) show that SI-GNNs successfully identify these edges, and Fig. 2(e–h) show that they also assign high weights to some irrelevant edges. Reasons are detailed in Lines 198–219.
> - Redundancy is Difficult to Eliminate: Figure 3 and Prop. 4.1 illustrate why tuning hyperparameters cannot eliminate redundancy; (2) Figure 4, Prop. 4.2, and Table 2 show the limitations of the +CL. Note that while EE helps mitigate its negative impact, it does not eliminate redundancy. Eliminating redundancy remains an open challenge.
> - Questions on Prop. 4.1: By definition, $H(Y|G_s^*)=0$. If $G_s^* \subseteq G_s$, we have $H(Y|G_s) = 0$. We cite Zhang et al. (2022) just to show that constraining the subgraph size (in theory) can be implemented via an additional loss term (in practice) -- please see Eqs. 12–14 in their paper.
>
> (2) Relationship Between SHD and AUC
>
> SHD and AUC evaluate different properties of explanations: SHD evaluates explanation inconsistency, while AUC evaluates explanation accuracy. There is no inherent relationship between the two metrics. Figure 6 just illustrates that the effectiveness of EE improves as $n$ increases.
>
> (3) Threshold Choice & Impact of Discretization on Redundancy
>
> We chose a threshold of 0.5 because determining whether an edge is important is essentially a binary classification task, and 0.5 is a standard and prior-free choice. In contrast, other thresholds or Top-K selections all require domain knowledge. Regarding redundancy, the choice of discretization method does have an effect, as it directly influences the number of retained edges, which in turn affects metric value such as SHD. But this does not change the core conclusion of our paper. This is because irrelevant edges might receive higher scores than truly important ones, which means that no matter how the threshold is set (as long as it ensures relevant edges are included), these irrelevant edges will always be retained. That said, we believe HS and ER, proposed in the ICLR 25 paper you suggested, are very important for the future design of SI-GNNs.
>
> (4) Empirical Evaluation of Deng & Shen’s Approach
>
> Table 2 in Appendix D provides empirical evidence that applying CL to Type I reduces both AUC and ACC. Other types yield similar results, which we can provide in the revised version if needed.
>
> (5) Relation to ICLR 25 paper
>
> Sorry, but we could not find the exact claim about ‘insufficient explanations’ in this paper. After reading it, we found that the most relevant aspect to our work is its discussion on the unreliability of existing FID metrics when redundancy exists (as they acknowledge redundancy might occur). This paper does not analyze why redundancy happens, nor does it provide clear evidence that the proposed strategies for improving faithfulness can address it. Given this, we believe our contributions remain distinct.
>
> We appreciate your time and feedback, and we hope our rebuttal addresses your concerns. We look forward to any further discussions if needed. Thank you!

---

> > ### Comment · Reviewer_6xUG · 2025-04-06
> >
> > Thank you for all clarifications.  I still have a few comments.
> >
> > - Evidence for redundancy: I understand the empirical evidence - but, again, I am skeptical that a well trained ensemble of SI-GNNs will identify a plausible explanation under confounding.  In hindsight, experiments on a confounded setting would help to figure out if the claim is solid or not.  Does this make sense to you?
> >
> > - Prop 4.1. $H(Y \mid G_s^*) = 0$ cannot follow from a definition: at the bare minimum it requires assuming that the underlying data generating progress is deterministic given the explanation (it some applications it might not be!). Would you agree?
> >
> > - Threshold: I see the argument that 0.5 is a natural threshold for *balanced* binary classification tasks, but I'm not sure that per-edge relevance prediction is a balanced task, especially for sparser explanations. Mind you, this is not a big issue for me, provided you clarify this aspect somewhere.
> >
> > **Post-rebuttal update**: I appreciate that the authors are willing to clarify the key issues I pointed out, so I will increase my score.  I don't think I will bump it further as, again, I am skeptical that (roughly speaking) ensembles of SI-GNNs can somehow work around spurious correlations/achieve plausibility.  One option would simply to drop claims in this direction, although doing so would -- I think -- substantially lessen the intended message of the paper.

---

> > > ### Author Response · Authors · 2025-04-07
> > >
> > > We sincerely appreciate your willingness to increase your score!!!
> > >
> > > **(1) Clarification on EE's Impact on Redundancy and Spurious Correlations**
> > >
> > > We completely agree with you that SI-GNNs + EE may not identify a plausible explanation under confounding conditions. We believe this limitation is not due to EE, but to the capability of SI-GNNs, which is a different research direction and not the focus of our work. EE is designed to mitigate the negative impact of redundancy, and the key contribution of our work is to raise awareness in the community that redundancy in explanations weakens explanation quality. Below, we present the reasons in detail.
> > >
> > > **Reasons:** In our paper, we argue that an explanation generated by SI-GNNs can be decomposed into two parts: (1) edges that the SI-GNN genuinely deems important, and (2) edges that are assigned high importance just because sufficient budget allocation (redundancy). If the second part causes the explanation inaccuracy, then EE can help improve AUC, because these edges generally exhibit high variance and tend to have lower average importance after EE.
> > >
> > > **What Happens if We Use Datasets with Spurious Correlations:** We ran experiments on SP-MOTIF (a synthetic dataset containing spurious correlations, proposed by Wu et al. (2022)) across all four types of SI-GNNs. Based on our results, we illustrate four instance-level cases. For simplicity, let $A$ represent a truly important edge, $B$ a spurious edge, and $C, D, E, F, \dots$ irrelevant edges.
> > > - **Case 1:** SI-GNN uses $A$ for classification. After EE, $A$ is retained, and other edges are given lower importance. AUC improves.
> > > - **Case 2:** SI-GNN uses $A$ for classification, with some use of $B$. After EE, both $A$ and $B$ are retained, and other edges are given lower importance. $A$ receives a higher score than $B$ due to more frequent occurrences. AUC improves.
> > > - **Case 3:** SI-GNN uses $B$ for classification, with some use of $A$. After EE, both $A$ and $B$ are retained, and other edges are given lower importance. $B$ receives a higher score than $A$ due to more frequent occurrences. AUC decreases.
> > > - **Case 4:** SI-GNN uses $B$ for classification. After EE, $B$ is retained, and other edges are given lower importance. AUC decreases.
> > >
> > > PS: (1) Given an instance, some edges that are neither $A$ nor $B$ may be consistently assigned high importance by the model. This does not affect our analysis, as shown in Case 2 in Figure 5. (2) In all cases, redundancy exists -- some irrelevant edges exhibit high variance across multiple runs. (3) Whether EE improves explanation accuracy depends on the SI-GNN’s capability and can vary across different models. (4) EE still provides valuable insights under confounding: after EE, we gain a clearer understanding of which spurious edges the model relies on for classification, as EE filters out the noise caused by redundancy. This helps researchers monitor and improve their algorithms more effectively.
> > >
> > > **Addressing Your Concerns on Spurious Correlations:** We guess your concern may stem from our discussion in Sec 3.2. In that section, we explore the potential reasons behind the explanation inconsistency observed in SI-GNNs. Zhang et al. (2023) suggest that spurious correlations cause explanation inconsistency. Our experiments reveal that when SI-GNNs are not affected by spurious correlations (in certain datasets), explanation inconsistency persists. This prompts us to investigate further, ultimately leading us to discover redundancy.
> > >
> > > **Summary:** We will revise Sec 3.2 to prevent any misunderstandings. We will include additional experiments and analyses in the revised version to clarify the limitations and effectiveness of EE under confounding. Furthermore, we will provide the necessary assumptions and clarifications regarding Prop 4.1 and the threshold, along with further discussion on both topics.
> > >
> > > If you have any further suggestions, we would greatly appreciate it if you could update your original review so that we can see them. We assure you that we will revise the paper according to your suggestions. Once again, we sincerely appreciate your constructive feedback and the opportunity to improve our work. Thank you :)

---

### Official Review · Reviewer_K3Wx · 2025-03-17

**Overall Recommendation:** 4

**Summary:**

This paper investigates the inconsistency in explanations generated by self-interpretable GNNs. It identifies redundancy—caused by weak conciseness constraints—as the root cause of explanation inconsistency, which in turn reduces trustworthiness. The paper argues that redundancy is difficult to eliminate completely but suggests a simple ensemble strategy to mitigate its effects. Extensive experiments across multiple datasets and models validate the claim that EE improves explanation consistency and accuracy.

**Claims And Evidence:**

Yes, the claims made in the submission are supported by clear and convincing evidence.

**Essential References Not Discussed:**

n/a

**Experimental Designs Or Analyses:**

I am not familiar with GNNs, so I can not judge it.

**Methods And Evaluation Criteria:**

Yes, the proposed method makes sense.

**Other Comments Or Suggestions:**

n/a

**Other Strengths And Weaknesses:**

S:
- The paper systematically examines why explanations of self-interpretable GNNs vary across runs with different random seeds. The investigation is thorough, combining theoretical analysis, extensive experiments, and case studies.

- The proposed Explanation Ensemble is a simple yet effective solution that requires no hyperparameter tuning and consistently improves explanation quality across various datasets and models.

W:
- The EE method requires training multiple models to aggregate explanations, which increases computational cost linearly. While it significantly improves consistency, this approach may not be feasible in resource-constrained settings.

**Questions For Authors:**

There are some other papers studying self-interpretable methods (although belong to NLP), would it be possible to discuss them in the related work?

[1] D-Separation for Causal Self-Explanation [2] Is the MMI Criterion Necessary for Interpretability? Degenerating Non-causal Features to Plain Noise for Self-Rationalization [3] Breaking Free from MMI: A New Frontier in Rationalization by Probing Input Utilization [4] MGR: Multi-generator Based Rationalization.

**Relation To Broader Scientific Literature:**

n/a

**Theoretical Claims:**

The paper makes two major theoretical claims:

- Redundancy is the primary cause of explanation inconsistency. The paper provides empirical evidence, showing that training instability and spurious correlations are not the main causes of inconsistency. Appendix A supports the argument that redundancy naturally arises due to weak conciseness constraints.

- Explanation Ensemble effectively mitigates redundancy. Empirical results consistently show EE improves explanation consistency and accuracy.

---

> ### Author Rebuttal · Authors · 2025-03-31
>
> Thank you for reviewing our work. We appreciate the references you suggested and will discuss them in the revised manuscript. Your recognition of our work truly means a lot to us.

---

### Decision · Program_Chairs · 2025-05-01

**Decision:**

Accept (poster)

**Comment:**

This paper presents the first systematic investigation into the trustworthiness of explanations generated by self-interpretable graph neural networks. It demonstrates that models trained with different initialization still recover the same meaningful features, while introducing some spurious ones. By aggregating, it is posible to improve the detection of relevant features.

The reviewers are all in agreement that this is valuable work, supported by solid evidence.